# Endosomal NOX2 oxidase exacerbates virus pathogenicity and is a target for antiviral therapy

Eunice E. To[1,2], Ross Vlahos[1], Raymond Luong[2], Michelle L. Halls[3], Patrick C. Reading[4], Paul T. King[5], Christopher Chan[2,6], Grant R. Drummond[7], Christopher G. Sobey[7], Brad R.S. Broughton[2], Malcolm R. Starkey[8], Renee van der Sluis[9], Sharon R. Lewin[9,10], Steven Bozinovski[1], Luke A.J. O'Neill[11], Tim Quach[12,13,14], Christopher J.H. Porter ⓘD [12,14], Doug A. Brooks[15], John J. O'Leary[16,17,18] & Stavros Selemidis[1,2]

The imminent threat of viral epidemics and pandemics dictates a need for therapeutic approaches that target viral pathology irrespective of the infecting strain. Reactive oxygen species are ancient processes that protect plants, fungi and animals against invading pathogens including bacteria. However, in mammals reactive oxygen species production paradoxically promotes virus pathogenicity by mechanisms not yet defined. Here we identify that the primary enzymatic source of reactive oxygen species, NOX2 oxidase, is activated by single stranded RNA and DNA viruses in endocytic compartments resulting in endosomal hydrogen peroxide generation, which suppresses antiviral and humoral signaling networks via modification of a unique, highly conserved cysteine residue (Cys98) on Toll-like receptor-7. Accordingly, targeted inhibition of endosomal reactive oxygen species production abrogates influenza A virus pathogenicity. We conclude that endosomal reactive oxygen species promote fundamental molecular mechanisms of viral pathogenicity, and the specific targeting of this pathogenic process with endosomal-targeted reactive oxygen species inhibitors has implications for the treatment of viral disease.

[1] Program in Chronic Infectious and Inflammatory Diseases, School of Health and Biomedical Sciences, College of Science, Engineering & Health, RMIT University, Bundoora, Victoria 3083, Australia. [2] Department of Pharmacology, Infection and Immunity Program, Biomedicine Discovery Institute, Monash University, Clayton, Victoria 3800, Australia. [3] Drug Discovery Biology, Monash Institute of Pharmaceutical Sciences, Monash University, Parkville, Victoria 3052, Australia. [4] Department of Microbiology and Immunology, The University of Melbourne, The Peter Doherty Institute for Infection and Immunity, Melbourne, Victoria 3000, Australia. [5] Monash Lung and Sleep, Department of Medicine, Monash Medical Centre, Monash University, Clayton, Victoria 3168, Australia. [6] Center for Systems Biology, Massachusetts General Hospital, Harvard Medical School, 185 Cambridge Street, Boston, Massachusetts 02114, USA. [7] Department of Physiology, Anatomy & Microbiology, School of Life Sciences, La Trobe University, Melbourne, Victoria 3086, Australia. [8] Priority Research Centre's Grow Up Well and Healthy Lungs, School of Biomedical Sciences and Pharmacy, Faculty of Health and Medicine, The University of Newcastle, and Hunter Medical Research Institute, New South Wales 2305, Australia. [9] The Peter Doherty Institute for Infection and Immunity, The University of Melbourne and Royal Melbourne Hospital, Melbourne, Victoria 3000, Australia. [10] Department of Infectious Diseases, Alfred Hospital and Monash University, Melbourne 3004, Australia. [11] School of Biochemistry and Immunology, Trinity Biomedical Sciences Institute, Trinity College Dublin, Dublin 2, Ireland. [12] ARC Centre of Excellence in Convergent Bio-Nano Science and Technology, Monash Institute of Pharmaceutical Sciences, Monash University, Parkville, Victoria 3052, Australia. [13] Medicinal Chemistry, Monash Institute of Pharmaceutical Sciences, Monash University, Parkville, Victoria 3052, Australia. [14] Drug Delivery Disposition and Dynamics, Monash Institute of Pharmaceutical Sciences, Monash University, Parkville, Victoria 3052, Australia. [15] School of Pharmacy and Medical Sciences, Sansom Institute for Health Research, Division of Health Sciences, University of South Australia, Adelaide 5001, Australia. [16] Discipline of Histopathology, School of Medicine, Trinity Translational Medicine Institute (TTMI), Trinity College Dublin, Ireland. [17] Sir Patrick Dun's Laboratory, Central Pathology Laboratory, St James's Hospital, Dublin 8, Ireland. [18] Molecular Pathology Laboratory, The Coombe Women and Infants University Hospital, Dublin 8, Ireland. Correspondence and requests for materials should be addressed to S.S. (email: Stavros.selemidis@RMIT.edu.au)

The production of reactive oxygen species (ROS) is a highly coordinated process achieved by enzymes of the NADPH oxidase (NOX) family. NOX enzymes are not present in prokaryotes but evolved ~1.5 billion years ago in single cell eukaroytes and are present in most eukaryotic groups including ameba, fungi, algae and plants, nemotodes, echinoderms, urochordates, insects, fish, reptiles and mammals[1, 2]. The functions of NADPH oxidases within eukaryotes are diverse, however, a common function is the generation of ROS by innate immune cells in response to pathogens. Indeed, orthologs of NADPH oxidase in plants (*ArtbohD* and *ArtbohF*), fungi (*NOXA/B*), and invertebrates *Caenorhabditis elegans* (Duox orthologs), *Drosophila melongaster* (*NOX5 homolog*, *d-NOX*, and *DUOX*) and mosquito *Aedes aegypti* (*NOXM* and *DUOX*) generate ROS with bactericidal activity that protects the host[1, 2]. Vertebrates including teleosts, amphibians, birds, and mammals possess a NOX2 NADPH oxidase that generates a burst of ROS within phagosomes to kill invading pathogens especially bacteria. However, the impact of ROS on virus infection is largely unknown.

ROS, such as superoxide anion and hydrogen peroxide ($H_2O_2$), are produced by mouse and human inflammatory cells in response to viral infection and enhance the pathology caused by viruses of low to high pathogenicity, including influenza A viruses[3–8]. The primary source of inflammatory cell ROS is the NOX2 oxidase enzyme[8–11]. Although NOX2 oxidase plays a role in the killing of bacteria and fungi via phagosomal ROS production, NOX2 oxidase does not appear to eliminate viruses in a manner analogous to that for bacteria. In fact, in the absence of NOX2, influenza A virus causes substantially less lung injury and dysfunction, and leads to lower viral burden suggesting that NOX2 oxidase-derived ROS promotes rather than inhibits viral infection[3–8]. However, it remains unknown how viruses cause ROS production and how these highly reactive oxygen molecules, which appear to be largely confined to their site of generation, contribute to disease.

After binding to the plasma membrane[12], viruses enter cells and ultimately endosomes by a variety of mechanisms resulting in viral RNA detection by endosomal pattern recognition receptors, including toll-like receptor 3 (TLR3), TLR7, and TLR9[13]. The specific receptor interaction depends upon either the Group (I–V) or genomic orientation (i.e., ssRNA, dsRNA or DNA) of the virus and triggers an immune response characterized by Type I IFN and IL-1β production, and B-cell-dependent antibody production[13]. Host nucleic acids and self-antigens are also detected by endosomal TLRs, and in autoimmune disease, mediate similar Type I IFN responses and stimulate antibody production against self-RNA and antigen. Notably, mice that are chronically deficient in NOX2 oxidase have an increased tendency to develop self-antibodies[14] and patients with chronic granulomatous disease, who have a defective capacity to generate ROS via the NOX2 oxidase, have elevated circulating Type I IFNs and auto-antibodies[15]. These observations are supportive of the notion that low levels of ROS result in an enhanced immune response. However, it remains unknown how ROS modulates inflammation and the pathology caused by viruses and whether targeted (and acute) abrogation of ROS may actually be beneficial in treating viral infection.

Here we hypothesize firstly, that the internalization of virus into endosomes results in ROS production and that this subdues essential immune pathways that would otherwise clear the virus; and secondly that the targeted inhibition of endosomal ROS markedly reduces viral pathogenicity. The identification of a mechanism to explain the paradoxical effect of ROS on viruses vs. other pathogens such as bacteria has the potential to facilitate the development of specific endosome-targeted antiviral therapies.

Our results demonstrate that, NOX2 oxidase is expressed in endosomes in mouse and human cells, and is activated following infection with a) single stranded RNA viruses irrespective of their classification including influenza A viruses, respiratory syncytial virus, rhinovirus, Dengue virus and HIV, as well as b) the DNA viruses vaccinia virus and herpes simplex virus. Activation of endosomal NOX2 is dependent on TLR7 for ssRNA viruses or TLR9 for DNA viruses and is the result of PKC activation. Endosomal NOX2 oxidase activity results in the spatially targeted generation of $H_2O_2$, which suppresses key antiviral and humoral signaling processes via the modification of a unique, highly conserved single cysteine residue (Cys98) on the ectodomain of TLR7. Accordingly, targeting endosomal ROS production with a NOX2 oxidase inhibitor suppresses influenza A virus patho-genicity. Finally these findings identify four conceptual advancements centered on ROS biology including: (1) identifi-cation of the endosome as a critical sub cellular compartment of ROS generation to virus infection, irrespective of strain; (2) molecular targets of ROS reside within the endosome revealing a paradigm in organelle-specific cell signaling; (3) endosomal ROS suppress antiviral signaling, and (4) endosome specific delivery of a ROS inhibitor is an effective treatment strategy for influenza viral disease.

## Results

**Influenza viruses drive endosomal ROS.** To address the potential role of endosomal ROS production in virus pathology we first focussed on influenza A viruses, which belong to the Group IV negative sense, ssRNA viruses of the *Orthomyxoviridae* family and are internalized by endocytosis. Exposure of mouse alveolar macrophages (AMs), mouse peritoneal RAW264.7 cells or bone marrow-derived macrophages (BMDMs) to influenza A virus strain HKx31 (H3N2) resulted in a dose and time-dependent increase in influenza nucleoprotein (NP) fluorescence (Supplementary Fig. 1a), which was almost abolished by the dynamin inhibitor, Dynasore (100 μM) indicating a clathrin-coated pit or caveolin-dependent mechanism of internalization (Supplementary Fig. 1b). Internalized virus displayed a strong co-location with the early endosomal marker EEA1 (Fig. 1a). However, not all of the NP was co-located with EEA1 indicating that influenza A virus was not present exclusively in early endosomes (Fig. 1a) and might have already entered late endosomes and/or lysosomes. NOX2 co-located with EEA1 in control and influenza infected cells (Fig. 1b, Supplementary Fig. 1c). Thus, the enzymatic machinery for ROS generation is present in early endosomes and this is significantly enhanced in influenza A virus infection, promoting co-localization with internalized virus.

Endosomal ROS production in response to viral uptake was assessed with OxyBURST[16]. Exposure to a series of low to high pathogenic seasonal and pandemic influenza A viruses resulted in rapid and dose-dependent increases in OxyBURST fluorescence in mouse primary AMs (Fig. 1c, d, Supplementary Fig. 2a, b, e, f) and human AMs (Fig. 1h). This OxyBURST-derived signal was abolished by addition of superoxide dismutase (SOD; 300 U/ml), which internalizes into the endosome along with the virus[17] and converts superoxide to $H_2O_2$ (Fig. 1e, f). In contrast the ROS signal was significantly increased in AMs from mice deficient in endosomal SOD (SOD3$^{-/-}$ mice), establishing the detection of a superoxide derivative (Supplementary Fig. 2c, d). For confirmation that ROS production occurred in acidified endosomes we demonstrated a co-location of OxyBURST fluorescence with LysoTracker (50 nM) in the presence of influenza virus (Fig. 1g). Inhibition of the vacuolar V-ATPase pump with bafilomycin A (100 nM), and thus inhibition of

endosomal acidification, abolished the LysoTracker fluorescence and endosomal ROS production in response to influenza A virus infection (Fig. 1g). Endosomal ROS was minimal in NOX2$^{-/y}$ AMs, but was unaffected in NOX4$^{-/-}$ macrophages and in macrophages treated with the NOX1 inhibitor ML171 (100 nM) (Fig. 1e, f, Supplementary Fig. 2h, i). Internalization of influenza

A virus into AMs was not impaired in NOX2$^{-/y}$ cells (Supplementary Fig. 2g), indicating that reduced endosomal ROS production was not due to reduced viral entry. In addition, heat- and UV-inactivated forms of influenza (replication-deficient) caused an increase in endosomal ROS production that was similar to the live virus control (Fig. 1i, j). Therefore,

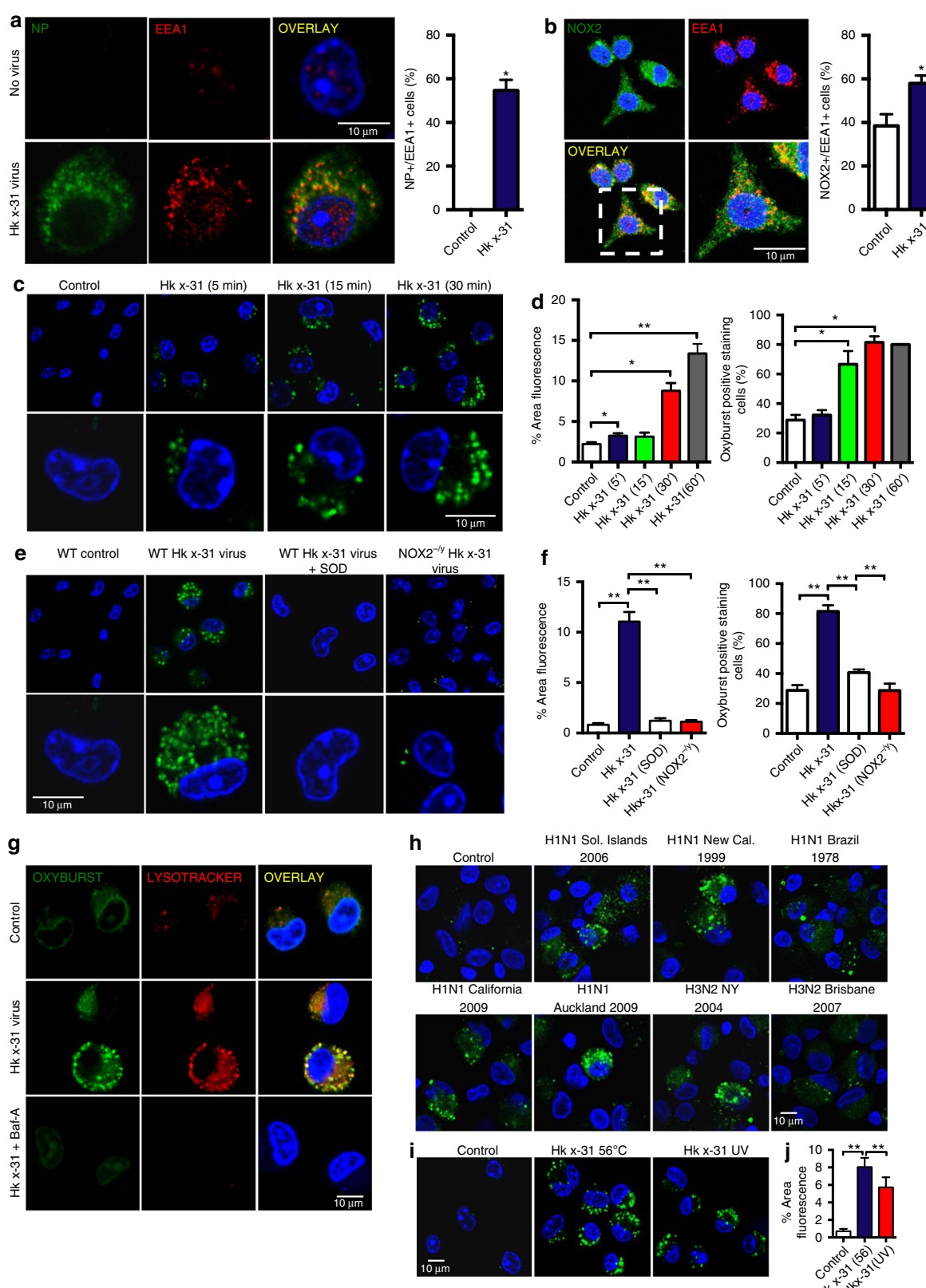

influenza A viruses, irrespective of subtype, strain and pathogenicity, stimulate NOX2, but not NOX4 or NOX1 oxidase-dependent ROS production in endosomes, and this involves endosomal acidification, but does not require viral replication.

**Endosome TLR7-NOX2 signaling axis.** RNA viruses are recognized by endosomal TLR7 (for ssRNA viruses)[18, 19] and TLR3 (dsRNA viruses), as well as the cytosolic sensors retinoic acid inducible gene I (RIG-I) (which can detect viral RNA bearing 5′ triphosphates[18] and NOD-like receptors (NLRs)[13, 20, 21]. We hypothesized that influenza A virus entry into acidified endosomes results in the liberation of viral RNA, activation of TLR7 and stimulation of NOX2 oxidase-dependent ROS production. Consistent with this suggestion, TLR7 co-locates with influenza A virus (Fig. 2a), NOX2 (Fig. 2b) and EEA1 (Fig. 2c, Supplementary Fig. 3a) and primary AMs from TLR7[−/−] mice, and TLR7- and MyD88-deficient BMDM, display minimal endosomal ROS production in response to influenza A virus (Fig. 2d, Supplementary Fig. 3b–e). The lack of endosomal ROS production in response to virus in TLR7[−/−] and MyD88[−/−] cells was not due to a reduced capacity of the NOX2 oxidase per se, as NOX2 activation by the PKC activator phorbol dibutyrate (PDB; $10^{-6}$ M) was similar in these cells and wild-type (WT) control cells (Supplementary Fig. 3c). As a second measure of NOX2 oxidase activity, we assessed enzyme assembly by examining the degree of association of the NOX2 catalytic subunit with the p47[phox] regulatory subunit. In unstimulated cells, there was very little co-localization of NOX2 and p47[phox] (Fig. 2e). However, influenza virus caused strong co-location of NOX2 and p47[phox], which was reduced by Dynasore or bafilomycin A pre-treatment, and almost abolished in TLR7[−/−] cells (Fig. 2e). To provide further evidence that the activation of TLR7 leads to endosomal ROS production, we used the specific TLR7 agonist, imiquimod (10 μg/ml). Imiquimod markedly increased endosomal ROS in AMs from human and WT mice, but not from NOX2[−/y] mice (Fig. 2f) or macrophages deficient in TLR7 or MyD88 (Supplementary Fig. 3b). Finally, we pulsed AMs or RAW264.7 cells with a guanidine- and uridine-rich ssRNA sequence (ssRNA40; 100 μM). In concentrations capable of increasing IL-1β, IL-6, and TNF-α mRNA via a TLR7-dependent mechanism (Supplementary Fig. 4), ssRNA40 caused elevated endosomal ROS production (Fig. 2g). In contrast, endosomal ROS production in response to influenza A virus was preserved in RIG-I[−/−], NLRP3[−/−], TLR2[−/−], and TLR4[−/−] macrophages, and in macrophages treated with the TLR3 inhibitor (50 μM) (Supplementary Fig. 3d–h).

We subsequently examined how TLR7 elicits the assembly and activation of endosomal NOX2 oxidase. NOX2 oxidase can be activated by protein kinase C, which triggers robust phosphorylation of key serine residues on p47[phox], resulting in a NOX2 oxidase-dependent oxidative burst[10]. To define the spatiotemporal regulation of PKC signaling and to assess its regulation by TLR7, we expressed the FRET biosensor cytoCKAR, to detect cytosolic PKC[22–24] in WT and TLR7[−/−] macrophages. The treatment of WT macrophages with influenza A virus or imiquimod elevated cytosolic PKC activity within 5 min, but this response was absent in TLR7[−/−] macrophages and in WT macrophages treated with Dynasore or bafilomycin A (Fig. 2h, i). A FRET biosensor method for cytosolic pERK1/2 activity[21, 24] showed that both influenza virus and imiquimod increased cytosolic pERK1/2 in a TLR7-dependent manner. In contrast, blocking pERK1/2 with PD98059 (30 μM) did not influence endosomal ROS production (Supplementary Fig. 5a, b) or the association of NOX2 with p47[phox] in response to influenza (Supplementary Fig. 5c). These data indicate that influenza A virus increases endosomal NOX2 oxidase activity via TLR7 and the downstream activation of PKC but not via pERK1/2.

We conclude that virus infection triggers a NOX2 oxidase-dependent production of ROS in endosomes using a process that is dependent on low pH. Indeed this conclusion is supported by the following experimental evidence. First it is known that reduced endosome acidification impairs the activation of TLR7 by viral RNA[18, 19]. Our study is in agreement with this finding, showing that NOX2 dependent ROS production to virus infection and to the TLR7 agonist imiquimod was abolished in TLR7[−/−] cells and also by pretreatment with bafilomycin A. Second, bafilomycin A suppressed PKC activation due to influenza virus and imiquimod treatment, and PKC is upstream of acute NOX2 activation[10, 11]. Third, bafilomycin A suppressed the association of p47phox-NOX2, which is a critical step for NOX2 assembly and activation.

**Viral strain independence of endosomal ROS.** Exposure of macrophages to rhinovirus (*picornaviridae, Group IV*), respiratory syncytial virus (*paramyxoviridae, Group V*), human parainfluenza virus (*paramyxoviridae, Group V*), human metapneumovirus (*paramyxoviridae, Group V*), Sendai virus (*paramyxoviridae, Group V*), Dengue virus (*flaviviridae Group IV*) or HIV (*retroviridae*, Group VI, ssRNA-RT virus) resulted in a significant elevation of endosomal ROS that was markedly suppressed in TLR7[−/−] macrophages, but unaffected in TLR9[−/−] cells (Fig. 3a, b). Both mumps virus (*paramyxoviridae Group V*) and Newcastle disease virus (NDV, *paramyxoviridae Group V*) failed to generate significant endosomal ROS (Fig. 3a, b), and it is noteworthy that these viruses primarily enter cells by a cell membrane fusion process and not via endocytosis. Rotavirus (rhesus monkey strain or bovine UK strain,

**Fig. 1** Seasonal and pandemic influenza A viruses induce endosomal ROS production via activation of NOX2 oxidase. **a, b** Confocal microscopy of wild-type (WT) mouse primary alveolar macrophages that were infected with influenza A virus strain HKx31 (MOI of 10) for 1 h and labeled with antibody to the early endosome antigen 1 (EEA1) and antibodies to either **a** influenza A virus nucleoprotein (NP) or **b** NOX2, and then with 4′,6′-diamidino-2-phenylindole (DAPI; blue). Also shown is the quantification of results (n = 5). **c, d** Time-dependent elevation in endosomal ROS levels in mouse primary alveolar macrophages as assessed by OxyBURST (100 μM) confocal fluorescence microscopy and labeled with DAPI (n = 5). **e, f** Endosomal ROS production in WT, NOX2[−/y] and superoxide dismutase (SOD; 300 U/ml)-treated WT mouse primary alveolar macrophages as assessed by OxyBURST confocal fluorescence microscopy in the absence or presence of HKx31 virus and labeled with DAPI (n = 5). **g** Uninfected and HKx31 virus-infected mouse primary alveolar macrophages were labeled with OxyBURST and the acidified endosome marker Lysotracker (50 nM). Some cells were treated with bafilomycin A (Baf-A; 100 nM) to suppress acidification of endosomes (n = 4). **h** Human alveolar macrophages infected with seasonal H3N2 (A/New York/55/2004, A/Brisbane/9/2007), seasonal H1N1 (A/New Caledonia/20/1999, A/Solomon Islands/3/2006) and pandemic A(H1N1) pdm09 strains (A/California/7/2009, A/Auckland/1/2009) and labeled with OxyBURST for endosomal ROS (n = 4). **i, j** Endosomal ROS production in WT mouse primary alveolar macrophages as assessed by OxyBURST fluorescence microscopy exposed to either heat (56 °C)-inactivated HKx-31 virus (to block virus fusion) or UV-inactivated HKx-31 virus (to block replication) and labeled with DAPI (n = 4). **a–i** Images are representative of >150 cells analyzed over each experiment. Original magnification ×100. **a, b, d, f** and **j** Data are represented as mean ± S.E.M. **a** and **b** Students' unpaired t-test *P < 0.05. **d, f** and **j** One-way ANOVA followed by Dunnett's post hoc test for multiple comparisons. *P < 0.05 and **P < 0.01. Scale bars: 10 μm

(*reoviridae Group III*)) exposure of macrophages also failed to generate endosomal ROS (Fig. 3a, b). The DNA viruses Herpes simplex virus 2 (*herpesviridae, Group I*) and vaccinia virus (*poxyviridae, Group I*) each caused an elevation in endosomal ROS in WT macrophages and TLR7$^{-/-}$ macrophages, but not in TLR9$^{-/-}$ macrophages (Fig. 3a, b). We conclude that the specific recognition of either ssRNA viruses by TLR7, or DNA viruses by TLR9, leads to a NOX2 oxidase-dependent burst of endosomal ROS.

**Bacteria and viruses activate distinct ROS pathways.** Plasma membrane TLRs, especially TLR1, TLR2, and TLR4, and not those present within endosomes (such as TLR7), sense bacteria resulting in the recruitment of mitochondria to macrophage phagosomes and mitochondrial dependent ROS production[25]. However, the stimulation of endosomal TLRs failed to augment mitochondrial ROS[25]. We confirmed that TLR7 activation with imiquimod, which caused a significant elevation in endosomal ROS (Fig. 2f), failed to increase macrophage mitochondrial

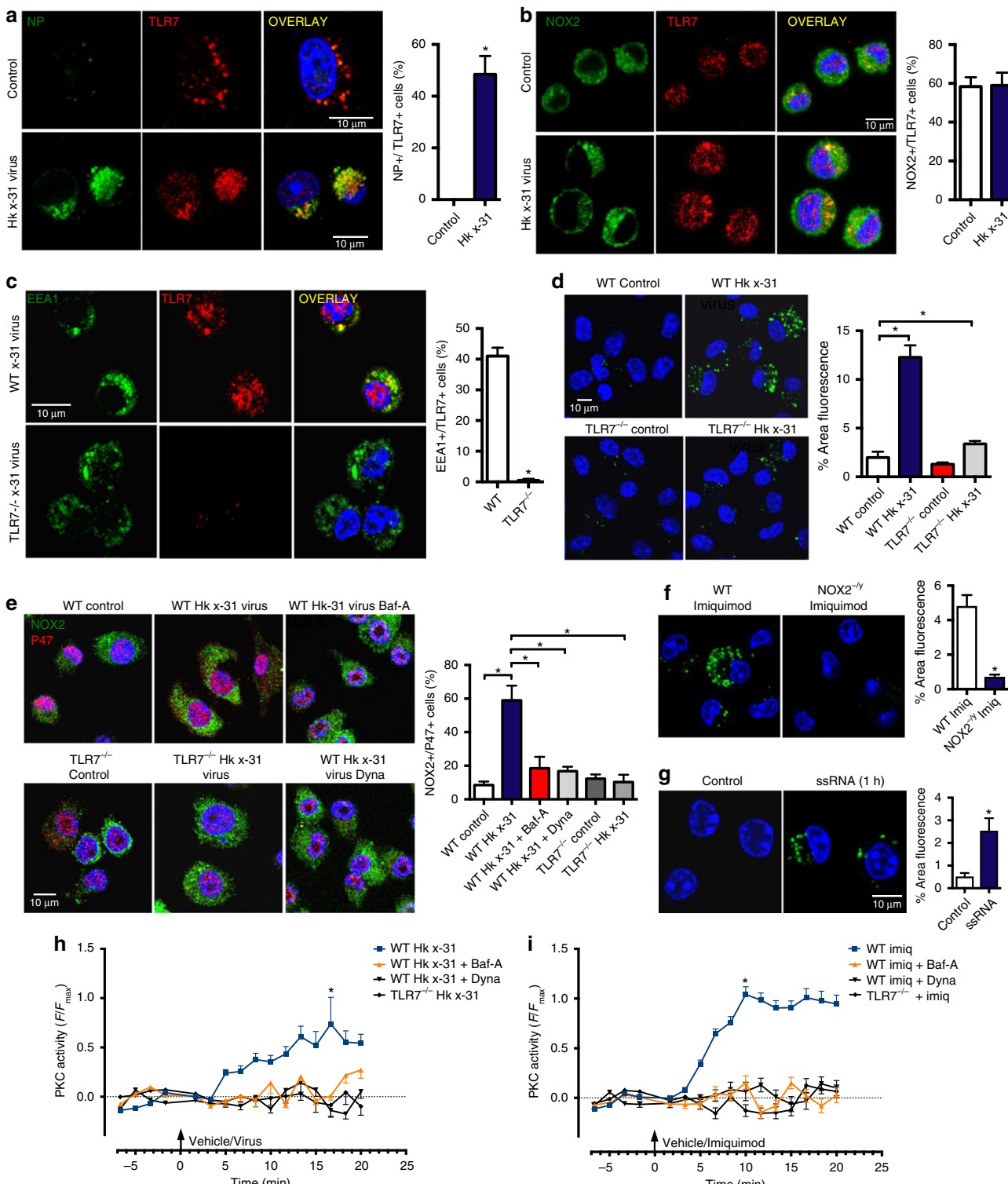

superoxide production (Supplementary Fig. 3i). We examined the production of phagosomal ROS in response to the Gram-positive bacteria *Streptococcus pneumoniae* (SP) or gram-negative non-typeable *Haemophilus influenzae* (NTHI). Both SP and NTHI caused ROS production in WT mouse macrophages (Fig. 4), which was enhanced in SOD3$^{-/-}$ cells (Supplementary Fig. 2j), but unaffected in TLR7$^{-/-}$ macrophages (Fig. 4). Thus, endosomal ROS production is not a characteristic of endocytosis per se, but a "pathogen (cargo)-specific" response. ROS produced for antibacterial purposes involves an obligatory role of mitochondria, which serves as a central hub to promote innate immune signaling. By contrast, ssRNA viruses do not employ these antibacterial ROS generating pathways.

**Endosomal $H_2O_2$ suppresses TLR7 immunity**. To establish the functional importance of endosomal ROS, we assessed the impact of NOX2 inhibition on the production of cytokines that are endosome TLR7-dependent and thus relevant to virus pathogenicity[19]. We first confirmed an endosome- and TLR7-dependent signal by showing that imiquimod caused a significant elevation in IFN-β, IL-1β, TNF-α, and IL-6 expression in WT macrophages, but not in TLR7$^{-/-}$ macrophages (Fig. 5a) or in macrophages treated with bafilomycin A (100 nM) (Fig. 5b). Second, pre-treatment with the NOX2 oxidase inhibitor and $H_2O_2$ scavenger, apocynin (300 μM) significantly enhanced IFN-β, IL-1β, TNF-α, and IL-6 expression in response to imiquimod, in WT macrophages but not in TLR7$^{-/-}$ macrophages, indicating that the suppressive effect of NOX2 oxidase-derived ROS on cytokine expression is dependent on TLR7 (Fig. 5a). In contrast, IFN-β, IL-1β, TNF-α, and IL-6 expression in response to the TLR3 agonist, poly I:C (25 μg/ml), was suppressed by apocynin pre-treatment (Supplementary Fig. 6a) whereas increases in these same cytokines triggered by the TLR9 agonist CpG (10 μg/ml), were unaffected by apocynin (Supplementary Fig. 6b). We further tested whether NOX2 oxidase influences TLR7 immunity in vivo. WT and NOX2$^{-/y}$ mice were treated with a single dose of imiquimod (50 μg per mouse, intranasally) for measurements of lung IFN-β, IL-1β, IL-6, and TNF-α after 24 h. This time point was chosen to reflect early phases of RNA infection. There were no discernible alterations in airway inflammation in response to imiquimod (Fig. 5c), however, imiquimod treatment resulted in elevated levels of IFN-β, IL-1β, IL-6, and TNF-α in NOX2$^{-/y}$ mice (Fig. 5d).

We sought to establish how endosomal NOX2 oxidase activity results in the suppression of TLR7-dependent responses and hypothesized that the parent species superoxide and its immediate downstream product, $H_2O_2$ are culprit mediators. Inactivation of superoxide by adding exogenous SOD (300 U/ml) failed to influence either basal or imiquimod-stimulated

expression of IFN-β, IL-1β, TNF-α, and IL-6, suggesting little role for superoxide itself in modulating TLR7 responses (Supplementary Fig. 7). To examine $H_2O_2$, we utilized catalase to inactivate the $H_2O_2$ generated within endosomes. We found that within 30 min, exposure to a FITC-labeled catalase resulted in co-localization with LysoTracker, confirming internalization into acidified endosomal compartments (Fig. 6a). A 1 h "pulse" exposure to catalase (1000 U/ml) resulted in significant elevations in IFN-β and IL-1β expression after 24 h in WT macrophages, but not in TLR7$^{-/-}$ macrophages (Fig. 6b). Moreover, imiquimod-dependent responses were significantly increased in the presence of catalase (Fig. 6c). The catalase-dependent increase in cytokines was significantly suppressed in WT macrophages treated with Dynasore (Fig. 6d) but unaffected in TLR2$^{-/-}$ macrophages (Fig. 6e, Supplementary Fig. 8a). The translocation of TLR7 to endosomes is governed by the actions of the chaperone protein, UNCB93. Indeed in the absence of UNCB93 there are substantial signaling defects due to the failure of the nucleotide-sensing TLRs to reach the endolysosomes, where they initiate MyD88/TRIF-dependent signaling pathways. In UNCB93$^{-/-}$ cells, the increase in cytokines to catalase treatment was significantly smaller than that observed in WT cells (Fig. 6f, Supplementary Fig. 8b). Thus, the suppressive actions of $H_2O_2$ are most likely occurring when TLR7 is located within the endosomal compartment. Catalase had no effect on TLR7, TREML4 or NLRP3 expression indicating that $H_2O_2$ does not modulate the expression of TLR7, a positive regulator of TLR7 activity (i.e., TREML4[26]) or NLRP3 that drives similar anti-viral cytokines to TLR7 (Fig. 6g–j). Therefore the effect of $H_2O_2$ is likely to be post-translational. We then examined whether endosomal NOX2 oxidase-derived $H_2O_2$ influences TLR7 responses in vivo. We administered catalase (1000 U per mouse) intranasally to WT mice and showed a three to four fold increase in lung IFN-β, IL-1β, TNF-α, and IL-6 after 24 h and this occurred prior to overt airway inflammation (Fig. 6k, l).

We questioned whether $H_2O_2$ released by endosomal NOX2 oxidase targets cysteine residues on protein domains of TLR7 that regulate receptor activity and are exposed upon activation within endosomal compartments[27]. These include Cys260, Cys263, Cys270, and Cys273 within the leucine repeat region as well as two additional cysteines, Cys98 and Cys445 that are unique to TLR7 (Supplementary Figs. 9 and 10). We performed site-directed mutagenesis to create a series of TLR7 mutants including (1) a mutant with all six of these cysteine residues replaced with alanine, (2) mutants with a dual mutation of Cys98 and Cys445 (TLR7$^{C98A/C445A}$), and (3) single mutations of Cys98 (TLR7$^{C98A}$) and Cys445 (TLR7$^{C445A}$). Transfection of WT TLR7 or TLR7$^{C445A}$ into TLR7$^{-/-}$ macrophages restored the ability of imiquimod to stimulate cytokine expression

**Fig. 2** Co-localization of TLR7 with influenza A virus, NOX2 and EEA1 is a signaling platform for endosomal ROS generation to influenza A virus via a TLR7 and PKC-dependent mechanism. **a–c** Confocal microscopy of mouse primary alveolar macrophages that were untreated or infected with influenza A virus HKx31 (MOI of 10) and labeled with antibodies to TLR7 and either **a** influenza A virus NP, **b** NOX2 or **c** EEA1, and then with 4′,6′-diamidino-2-phenylindole (DAPI). Quantification data from multiple experiments are also shown (n = 5). **d** Endosomal ROS production in WT and TLR7$^{-/-}$ mouse primary alveolar macrophages as assessed by Oxyburst (100 μM) fluorescence microscopy in the absence or presence of HKx31 virus and labeled with DAPI (n = 6). **e** Immunofluorescence microscopy for assessment of NOX2 and p47phox association. WT and TLR7$^{-/-}$ immortalized bone marrow-derived macrophages (BMDMs) were untreated or infected with HKx31 virus, (MOI of 10) in the absence or presence of bafilomycin A (Baf-A; 100 nM) or dynasore (Dyna; 100 μM), and then labeled with antibodies to NOX2 and p47phox. Also shown is the quantification of the results (n = 5). **f, g** Endosomal ROS production in WT and NOX2$^{-/y}$ mouse primary alveolar macrophages as assessed by Oxyburst fluorescence microscopy in the absence or presence of **f** imiquimod (Imiq; 10 μg/ml) and **g** ssRNA (100 μg/ml) and co-labeled with DAPI. (n = 5). **h, i** Cytosolic PKC activity as assessed by FRET analysis in WT and TLR7$^{-/-}$ BMDMs. Cells were either treated with vehicle controls or with bafilomycin A (100 nM) or dynasore (10 μM) and then exposed for 25 min to influenza A virus (HKx31, MOI of 10) or imiquimod (10 μg/ml) (n = 3). **a–g** Images are representative of >150 cells analyzed over each experiment. Original magnification ×100. All data are represented as mean ± S.E.M. **a, b, c, f** and **g** Student's unpaired *t*-test *P < 0.05. **d, e, h** and **i** One-way ANOVA followed by Dunnett's post hoc test for multiple comparisons. *P < 0.05. Scale bars: 10 μm

in these cells; however, transfection with the TLR7 containing the 6 mutations, the $TLR7^{C98A/C445A}$ or the $TLR7^{C98A}$ did not (Fig. 7a). Catalase (1000 U/ml) treatment had little or no effect on cytokine expression in cells expressing the mutated TLR7, $TLR7^{C98A/C445A}$ or $TLR7^{C98A}$ whereas it markedly increased

cytokine expression in cells with WT TLR7 or $TLR7^{C445A}$ (Fig. 7a).

We performed sequence analysis using both multiple sequence analysis algorithms (i.e., CLUSTAL OMEGA) and pair-wise sequence analysis (NCBI, Blast) with human TLR7 as a reference

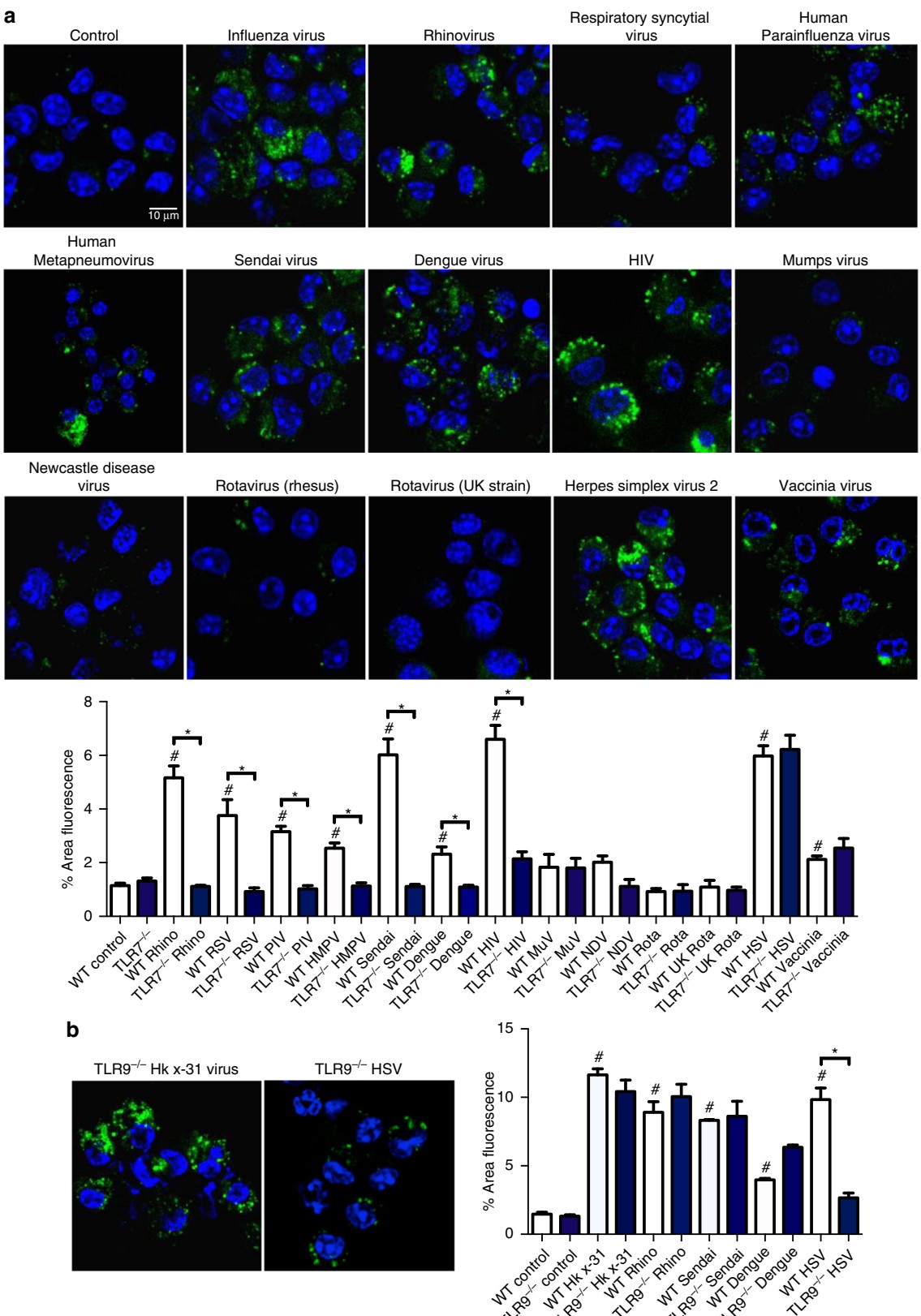

point. Using the multiple sequence analysis we identified that Cys98 was unique to TLR7 and fully conserved in vertebrate TLR7 including from teleosts to man (Fig. 7b, Supplementary Figs. 9 and 10). Pair-wise sequence alignment showed that Cys98 was the only cysteine residue of the 27 cysteines on TLR7 that was unique to TLR7 and fully conserved in vertebrates (Supplementary Table 1). We suggest that $H_2O_2$ produced by endosomal NOX2 oxidase is likely to modify a single and evolutionary conserved unique cysteine residue i.e., Cys98 located on the endosomal face of TLR7, resulting in a dampened antiviral cytokine response. Potentially this signifies Cys98 of TLR7 as a novel redox sensor that controls immune function during viral infections. The precise type of molecular modification of this cysteine by $H_2O_2$, however, is currently unknown and certainly warrants further investigation.

**NOX2 oxidase dampens antibody production**. We examined if the suppressive effect of endosomal NOX2 oxidase activity on Type I IFN and IL-1β expression also occurs following influenza A virus infection. Firstly, virus triggered translocation of the transcription factor, IRF-7, to the nucleus of WT BMDMs, but not TLR7$^{-/-}$ BMDMs, indicating that influenza A virus activates TLR7-dependent antiviral signaling in macrophages (Supplementary Fig. 11). Second, virus elevated IFN-β, IL-1β, IL-6, and TNF-α expression to a greater extent in NOX2$^{-/y}$ AMs (Fig. 8a). Third, influenza A virus (Hkx31; $10^5$ PFU per mouse) infection in mice in vivo for 24 h resulted in greater increases in lung IFN-β, IL-1β, IL-6, and TNF-α mRNA (Fig. 8b), as well as serum (Fig. 8c) and lung IFN-β protein (Fig. 8d) in NOX2$^{-/y}$ mice. Thus a fully functional NOX2 oxidase suppresses anti-viral cytokine production triggered by influenza A virus.

TLR7 is essential for the activation of B-cells and for antibody production. To test whether NOX2 oxidase suppresses TLR7-dependent immunity to influenza A virus in vivo, we used heat-inactivated, replication-deficient influenza A virus as a stimulus, and hence a form of virus expected to mainly trigger engagement of the TLR7 PRR with very little contribution of RIG-I and NLRP3[20]. Intranasal inoculation with inactivated virus had no effect on weight loss over 7-days (Fig. 8e) or airways BALF inflammation (Fig. 8f). NOX2 deletion resulted in a significant elevation in lung levels of IFN-β and IL-1β (Fig. 8g), and in both serum and BALF levels of IgA, total IgG, IgG1, IgG2b, and IgG3 (Fig. 8h, i). Therefore, activation of endosomal NOX2 oxidase following influenza A virus infection results in the suppression of antiviral cytokines and humoral immunity via the suppression of antibody production—processes that are required for optimal clearance of the virus and resistance to re-infection.

**Endosomal targeted NOX2 inhibitor**. We synthesized an innovative molecular targeting system, to deliver a specific NOX2 oxidase inhibitor (i.e., gp91ds-TAT) directly to endosomes, so as to disrupt the viral signaling platform by abrogating ROS production. To do this we generated a tripartite structure comprising gp91ds-tat conjugated to the membrane anchor cholestanol via a PEG-linker at the N-terminal region of the peptide. Similar constructs have been shown previously to enhance endosome localization for inhibitors of the enzyme beta secretase[28]. A Cy5 fluorophore conjugated to cholestanol using the same PEG linker resulted in cytosolic fluorescence in the peri-nuclear region and co-localization with EEA1, NOX2 and influenza virus NP following viral infection in a dynasore (100 μM)-sensitive manner providing evidence for endocytosis as its primary mode of cell entry (Fig. 9a–e). Superoxide generation in macrophages in vitro was suppressed with at least a 10-fold greater potency by cholestanol-conjugated gp91ds-TAT (Cgp91ds-TAT) when compared to the unconjugated drug (Ugp91ds-TAT; Fig. 9f), which is not attributed to enhanced ROS scavenging properties of the compound (Fig. 9g).

We examined whether Cgp91ds-TAT suppresses disease severity following influenza A virus infection in vivo. Daily intranasal administration of Cgp91ds-TAT (0.02 mg/kg/day) from 1 day prior, until day 3 post-influenza A virus infection resulted in a ~40% reduction in airways inflammation (Fig. 9h), whereas Ugp91ds-TAT had no significant effect (Fig. 9h). Cgp91ds-TAT significantly increased lung Type I IFN-β mRNA levels compared to the control virus group, whereas Ugp91ds-TAT failed to do so (Fig. 9i). To eliminate the possibility that this improvement in NOX2 inhibition by cholestanol conjugation of gp91ds-TAT was attributed to cholestanol-PEG linker per se, we conjugated the cholestanol PEG-linker to a scrambled gp91ds-TAT (Sgp91ds-TAT) and examined its effect against influenza infection in vivo. Sgp91ds-TAT had no effect on airway inflammation, lung IFN-β mRNA levels and superoxide production (Supplementary Fig. 12). Increasing the dose of the Ugp91ds-TAT by 10-fold to 0.2 mg/kg/day significantly reduced the weight loss caused by influenza A virus at day 3 and almost abolished airway inflammation, as well superoxide production in BALF inflammatory cells, similar to Cgp91ds-TAT at the same dose (Fig. 9j–l). Strikingly, both Cgp91ds-TAT (0.2 mg/kg/day) and Ugp91ds-TAT (0.2 mg/kg/day) caused an almost 10,000-fold, decrease in lung influenza A viral burden (Fig. 9m). Thus, suppression of endosome NOX2 oxidase via nasal administration of gp91ds-TAT results in a substantial reduction in influenza A virus pathogenicity. This is an innovative approach for suppressing NOX2 oxidase activity that occurs within the endosome compartment. We would like to emphasize that our custom-made NOX2 inhibitor is unlikely to solely suppress endosome NOX2. However, our inhibitor is specifically and preferentially delivered via the endocytic compartment owing to the cholestanol conjugation. In support of this, our findings of Fig. 9a and b show that cholestanol conjugation results in a drug delivery system that promotes endosome delivery i.e our drug displayed a strong degree of co-location with EEA1$^+$ endosomes that was abolished by dynasore pretreatment. This delivery system brings a NOX2 inhibitor to the predominant site of action that relates to virus infection (see Fig. 9d showing strong co-location of viral nucleoprotein and our NOX2 inhibitor).

**Fig. 3** Endosomal ROS production to ssRNA and DNA viruses are via TLR7 and TLR9-dependent mechanisms, respectively. **a** Endosomal ROS production in WT and TLR7$^{-/-}$ bone marrow-derived macrophages as assessed by OxyBURST (100 μM) fluorescence microscopy in the absence or presence of influenza A virus (HKx31 virus), rhinovirus (rhino), respiratory synctitial virus (RSV), human parainfluenza virus (PIV), human metapneumovirus (HMPV), sendai virus, dengue virus, human immunodeficiency virus (HIV), mumps virus (MuV), Newcastle disease virus (NDV), rotavirus (UK and bovine strains), herpes simplex virus 2 (HSV-2), and vaccinia virus and labeled with 4',6'-diamidino-2-phenylindole (DAPI). Also shown is the quantification of the results ($n = 5$). **b** Endosomal ROS production in WT and TLR9$^{-/-}$ mouse primary alveolar macrophages as assessed by OxyBURST fluorescence microscopy in the absence or presence of HKx31 virus, rhinovirus, sendai virus, dengue virus, and herpes simplex virus 2 (HSV-2) and labeled with DAPI ($n = 5$). **a** and **b** Images are representative of >150 cells analyzed over each experiment. Original magnification ×100. All data are represented as mean ± S.E.M. One-way ANOVA followed by Dunnett's post hoc test for multiple comparisons. $^\#P < 0.05$ compared to WT control. $^*P < 0.05$ comparisons indicated by horizontal bars. Scale bars: 10 μm

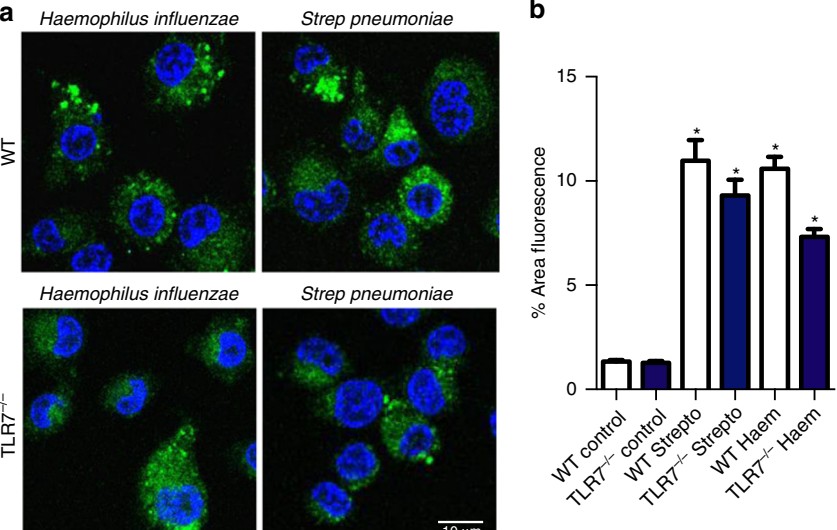

**Fig. 4** Bacteria-induced ROS production is distinct from virus-dependent ROS mechanisms. **a** Phagosomal superoxide production to *Haemophilus influenzae* and *Streptococcus pneumoniae* as assessed by OxyBURST (100 μM) fluorescence microscopy in WT and TLR7$^{-/-}$ immortalized bone marrow derived macrophages. Images are representative of >150 cells analyzed over each experiment. Original magnification ×100. **b** Is the quantification of the results ($n = 5$). All data are represented as mean ± S.E.M. One-way ANOVA followed by Dunnett's post hoc test for multiple comparisons. *$P < 0.05$ compared to WT control. Scale bar: 10 μm

We speculate that following internalization into the endosome the drug is most likely on the luminal face of the endosome membrane and due to the TAT portion can penetrate the membrane and suppress NOX2 activity. The drug might still be able to diffuse towards other sites or locations of NOX2, however, we believe the immediate and primary site of action will be NOX2 activity at the endosome, given that the drug appears to be selectively delivered via the endocytic pathway.

## Discussion

We have accumulated evidence that virus entry into endosomal compartments triggers a NOX2 oxidase- dependent production of ROS in endosomes. However, some ROS may have possibly been generated in sites outside of the endosome, which may have then diffused into the endosomes. As such the site of ROS generation and site of detection in some instances might be distinct. We suggest that the major contributor to endosomal concentrations of superoxide will be superoxide generated directly in this compartment. Superoxide is the primary product of NOX2 and it will only be generated within the endosome compartment owing to the topology of the NOX2 and the unidirectional transfer of electrons through this catalytic subunit. In keeping with this, it is well regarded that superoxide does not travel far from its site of generation due to its negative charge. By contrast to superoxide, H$_2$O$_2$ has some capacity to permeate membranes and diffuse, and as such, it can be envisaged that some endosome H$_2$O$_2$ might have been generated elsewhere by NOX2 expressed in other sites of the cell such as the plasma membrane. However for several lines of evidence we suggest that it is very likely that little remotely generated H$_2$O$_2$ is finding its way into the endosome compartment. First we demonstrate that PKC activation following virus infection, which is critical for NOX2 activation, is significantly impaired if: (1) the virus is prevented from entering cells (Fig. 2h, i); (2) endosome acidification is blocked by Bafilomycin A (Fig. 2h, i) or (3) if TLR7 is absent (i.e., TLR7$^{-/-}$ macrophages are used). Therefore, endosomal NOX2 derived ROS generation occurs only after virus has entered endosomes and activates endosome-specific pathways, lending further credence to endosome NOX2 as the predominant site of H$_2$O$_2$ generation. Second, elegant studies addressing spatialtemporal aspects of H$_2$O$_2$ diffusion clearly demonstrate that H$_2$O$_2$ diffusion in the cytoplasm is strongly limited, but is instead localized to near the sites of its generation[29], providing evidence that endosome H$_2$O$_2$ is unlikely to have been generated at a remote site.

Here we demonstrate that endosomal ROS are essential negative regulators of a fundamental molecular mechanism of viral pathogenicity that impacts on antiviral immunity and the capacity of the host to fight and clear viral infections. Importantly, this effect is conserved, regardless of viral classification, for all viruses that enter cells via the endocytic pathway, and is TLR7 dependent. This provides a potential target for antiviral therapy for a range of viruses that cause significant morbidity and mortality worldwide. Previous work has demonstrated that a deficiency in the phagocyte NADPH oxidase can result in exacerbated inflammation and necrosis in response to fungal components including β-glucans that activate TLR2 and dectin-1[30]. Our study is not a universal mechanism that defines how NOX2 oxidase might be regulating inflammation to fungi and other pathogens but certainly provides strong mechanistic insight into ROS-dependent regulation of antiviral immunity.

Intriguingly, our study raises a broader paradox: Why does a mammalian cell generate ROS that may ultimately cause harm (i.e., by promoting viral pathogenicity)? We hypothesize that suppression of TLR7 activation by endosomal NOX2 is a hitherto unrecognized mechanism that has evolved to inhibit an inflammatory response against self-RNA/antigens and the development of autoimmunity, but which in a very similar manner results in a host response to viral RNA that inadvertently exacerbates viral pathogenicity. Realizing the delicate balance between viral clearance and the induction of an autoimmune response, the current data suggest the potential to employ short term suppression of endosomal ROS as a means of reducing viral pathogenicity without causing long term problems with autoimmunity.

## Methods

**Viruses**. The influenza A virus vaccine strains HKx31 (H3N2) and BJx109 (H3N2) were kindly provided by A/Prof John Stambas (School of Medicine, Deakin

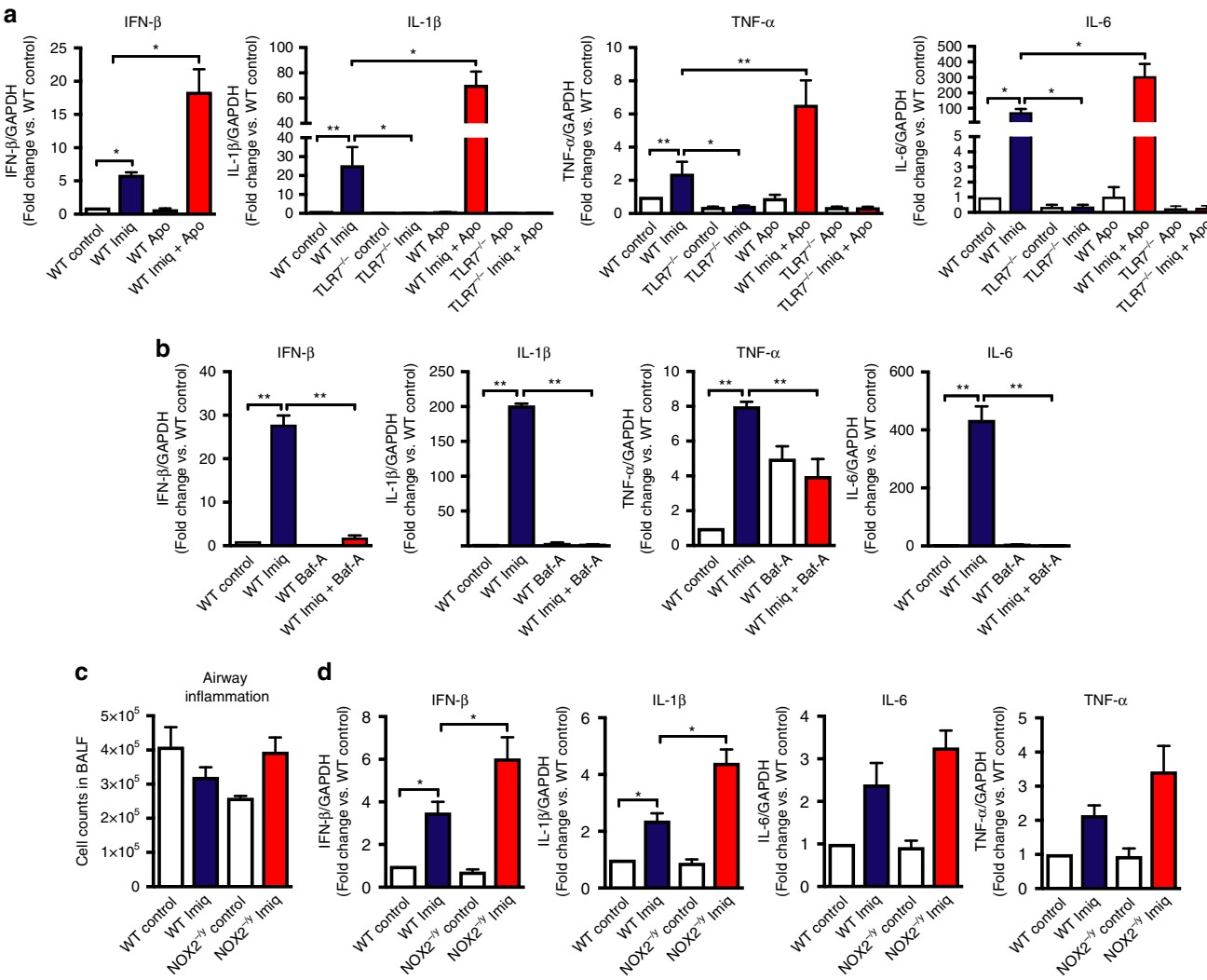

**Fig. 5** Endosomal NOX2 oxidase suppresses cytokine expression in response to TLR7 activation in vitro and in vivo. **a**, **b** WT and TLR7$^{-/-}$ immortalized bone marrow-derived macrophages (BMDMs) were untreated or treated with imiquimod (Imiq; 10 μg/ml) in the absence or presence of **a** apocynin (Apo; 300 μM) or **b** bafilomycin A (Baf-A; 100 nM), and IFN-β, IL-1β, TNF-α and IL-6 mRNA expression determined by QPCR after 24 h ($n = 6$). **c**, **d** WT and NOX2$^{-/y}$ mice were administered with imiquimod (50 μg per mouse, intranasal) and **c** total airway inflammation quantified by bronchoalveolar lavage fluid analysis and **d** cytokine expression assessed 24 h later ($n = 5$). **a**, **b**, **d** Responses are relative to GAPDH and then expressed as a fold-change above WT controls. **a**–**d** Data are represented as mean ± S.E.M. **a**, **b** and **d** Kruskal–Wallis test with Dunn's post hoc for multiple comparisons. **c** One-way ANOVA followed by Dunnett's post hoc test for multiple comparisons. Statistical significance was accepted when $P < 0.05$. *$P < 0.05$; **$P < 0.01$

University, CSIRO) and A/Prof. Patrick Reading (Department of Immunology and Microbiology, University of Melbourne, The Peter Doherty Institute for Infection and Immunity). Human strains of influenza A virus, including seasonal H3N2 (A/New York/55/2004, A/Brisbane/9/2007), seasonal H1N1 (A/Brazil/11/1978, A/New Caledonia/20/1999, A/Solomon Islands/3/2006), A(H1N1)pdm09 strains (A/California/7/2009, A/Auckland/1/2009), rhinovirus (RV16 strain), respiratory syncytial virus (strain A2), human parainfluenza virus type-3 (C243), human metapneumovirus (strain CAN97-83), mumps virus (strain Enders), and Newcastle disease virus (strain V4) were provided by A/Prof Patrick Reading. Additional viruses were provided by the following people: dengue virus serotype 2 (Vietnam 2005 isolate, Associate Prof Elizabeth McGraw; Monash University); rotavirus (Rhesus and UK strains, A/Prof Barbara Coulson, Department of Microbiology and Immunology, The Peter Doherty Institute for Infection and Immunity); sendai virus (Cantell strain, Dr Ashley Mansell from Hudson Institute of Medical Research, Monash University), herpes simplex virus type-2 (strain 186; Dr Niahm Mangan, Hudson Institute of Medical Research, Monash University), vaccinia virus (Western Reserve strain, WR NIH-TC; A/Prof. David Tscharke, Australia National University), and HIV (NL4-3(AD8)-EGFP strain, Prof. Sharon Lewin, The Peter Doherty Institute for Infection and Immunity, The University of Melbourne). The viruses were provided in phosphate buffered saline (PBS, Cat # D8537, Sigma, USA) and stored at −80 °C until used. On the day of use, virus was thawed quickly and incubated at 37 °C prior to infection. Where indicated, HKx31 virus was inactivated by heat (56 °C) for 30 min or UV light (30 min).

**Bacteria**. *S. pneumoniae* EF3030 (capsular type 19 F) was used as the parent *S. pneumoniae* strain in all experiments (provided by Dr. Odilia Wijburg, University of Melbourne, Australia). *S. pneumoniae* EF3030 is a clinical isolate that is frequently used as a model of human carriage as it typically colonizes the nasopharynx in the absence of bacteremia. For infection experiments, pneumococci were grown statically at 37 °C in Todd-Hewitt broth, supplemented with 0.5% yeast extract, to an optical density (600 nm) of 0.4–0.45. Cultures were placed on wet ice for 5 min and frozen in 8% (v/v) glycerol at −70 °C. Live bacterial counts were confirmed prior to each experiment. A defined strain of non-typeable *H. influenzae* (NTHi; MU/MMC-1) was previously typed and sequenced and demonstrated to be NTHi, as we have previously shown[31].

**Conjugation of NOX2 oxidase inhibitors**. Preparation of gp91 ds-tat (YGRKK-RRQRR-RCSTR-IRRQL-NH$_2$) was carried out by standard Fmoc solid-phase peptide synthesis (SPPS) on Fmoc-PAL-PEG-PS resin (Life Technologies, USA, loading 0.17 mmol/g). Fmoc deprotection reactions were carried out using 20% v/v piperidine in *N,N*-dimethylformamide (DMF). Coupling reactions were carried out using Fmoc-protected amino acids with *O*-(6-chlorobenzotriazol-1-yl)-*N,N,N′,N′*-tetramethyluronium hexafluorophosphate (HCTU) as coupling agent and *N,N*-diisopropylethylamine (DIPEA) as activating agent. Reactions were monitored using the 2,4,6-trinitrobenzenesulfonic acid test to indicate the absence or presence of free amino groups. The alternating

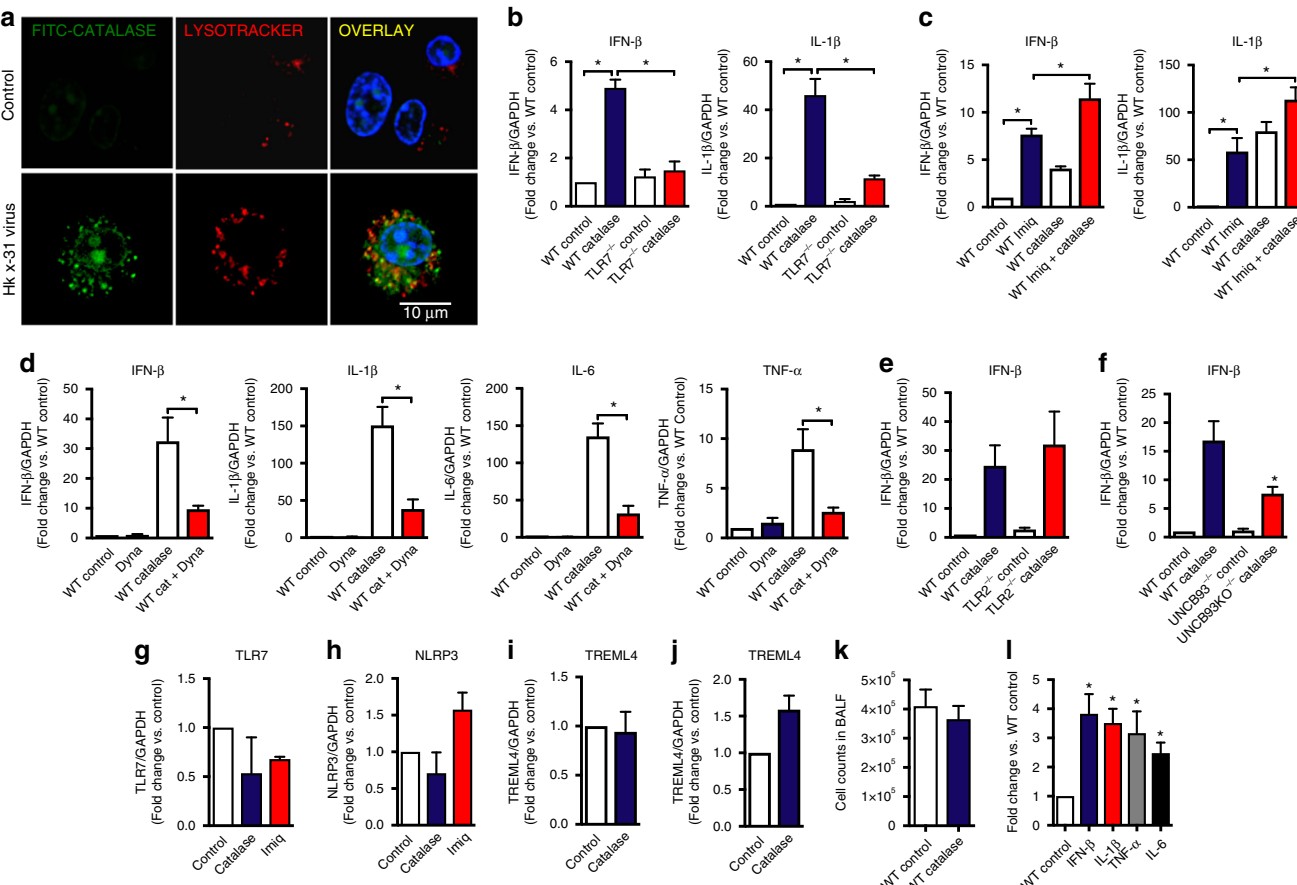

**Fig. 6** Endosomal NOX2 oxidase-derived hydrogen peroxide (H$_2$O$_2$) inhibits cytokine expression in response to TLR7 activation in vitro and in vivo. **a** WT mouse primary alveolar macrophages were either left untreated or treated with FITC-labeled catalase for 5 min prior to infection with HKx31 virus (MOI of 10). Cells were labeled for Lysotracker (50 nM) and colocalization of Lysotracker and FITC catalase assessed by confocal microscopy. Images are representative of >100 cells analyzed over each experiment. Original magnification ×100 ($n = 3$). **b** WT and TLR7$^{-/-}$ immortalized bone marrow-derived macrophages (BMDMs) were left untreated or treated for 1 h with catalase (1000 U/ml) and IFN-β and IL-1β, mRNA expression determined by QPCR after 24 h ($n = 7$). **c** WT BMDMs were left untreated or treated for 1 h with imiquimod (Imiq) in the absence or presence of catalase (1000 U/ml), IFN-β and IL-1β, mRNA expression assessed 24 h later by QPCR ($n = 6$). **d** WT BMDMs were treated for 30 min with either DMSO (0.1%) or dynasore (Dyna; 100 μM) and then with catalase (1000 U/ml) for 1 h. Cytokine mRNA expression determined by QPCR after 24 h ($n = 6$). **e** WT and TLR2$^{-/-}$ immortalized BMDMs were treated with catalase (1000 U/ml) for 1 h and cytokine mRNA expression determined by QPCR after 24 h ($n = 6$). **f** WT and UNCB93$^{-/-}$ immortalized BMDMs were treated with catalase (1000 U/ml) for 1 h and cytokine mRNA expression determined by QPCR after 24 h ($n = 6$). **g–i** WT BMDMs were treated for 1 h with either catalase or imiquimod (10 μg/ml) and **g** TLR7, **h** NLRP3 or **i** TREML4 mRNA expression determined by QPCR after 24 h ($n = 6$). **j** Mice were intranasally treated with catalase (1000 U per mouse) and then lung expression of TREML4 was determined by QPCR ($n = 5$). **k** and **l** Catalase (1000 U per mouse, intranasal) was administered to WT mice and **k** total BALF airway inflammation and **l** lung cytokine expression assessed 24 h later ($n = 5$). **b–j** and **l** Responses are relative to GAPDH and then expressed as a fold-change above WT controls. **b–h** and **l** Kruskal–Wallis test with Dunn's post hoc for multiple comparisons. **i** and **j** Mann–Whitney Wilcoxon test. All data are represented as mean ± S.E.M. Statistical significance was taken when the $P < 0.05$. *$P < 0.05$. Scale bar: 10 μm

sequence of deprotection and coupling reactions was carried out manually for all 20 amino acid residues using the appropriate Fmoc- and side-chain protected amino acids. After a final de-protection step, a small portion of the peptide was cleaved from resin using trifluoroacetic acid (TFA)/triisopropylsilane (TIPS)/1,2-ethanedithiol (EDT)/water (92.5:2.5:2.5:2.5) for 4 h, during which time the side-chain protecting groups were simultaneously removed. The crude peptide was then purified by reverse-phase high-pressure liquid chromatography using a Phenomenex Luna 5 C8 (2) 100 Å AXIA column (10 Å, 250 × 21.2 mm) with 0.1% TFA/water and 0.1% TFA/ACN as the buffer solutions. The purified gp91 ds-tat peptide was confirmed as having the correct molecular weight by ESI-MS analysis: calcd. for C$_{109}$H$_{207}$N$_{52}$O$_{25}$S [M + 5H$^+$] $m/z$ 535.3, obs. $m/z$ 535.7; calcd. for C$_{109}$H$_{208}$N$_{52}$O$_{25}$S [M + 6H$^+$] $m/z$ 446.3, obs. $m/z$ 446.6; calcd. for C$_{109}$H$_{209}$N$_{52}$O$_{25}$S [M + 7H$^+$] $m/z$ 382.7, obs. $m/z$ 382.9.

Preparation of cholestanol-conjugated gp91 ds-tat (cgp91 ds-tat; Ac-Asp(OChol)-PEG4-PEG3-PEG4-gp91-NH$_2$) was carried out by manual SPPS from resin-bound gp91 ds-tat (as described above), using Fmoc-PEG4-OH, Fmoc-PEG3-OH, Fmoc-PEG4-OH and Fmoc-Asp(OChol)-OH as the amino acids. After the final deprotection step, the $N$-terminus was capped using a mixture of acetic anhydride and DIPEA in DMF and the peptide construct was cleaved from resin using TFA/TIPS/EDT/water (92.5:2.5:2.5:2.5). The crude peptide was purified

as described previously to give cgp91 ds-tat: calcd. for C$_{173}$H$_{319}$N$_{56}$O$_{43}$S [M + 5H$^+$] $m/z$ 780.3, obs. $m/z$ 780.6; calcd. for C$_{173}$H$_{320}$N$_{56}$O$_{43}$S [M + 6H$^+$] $m/z$ 650.4, obs. $m/z$ 650.7; calcd. for C$_{173}$H$_{321}$N$_{56}$O$_{43}$S [M + 7H$^+$] $m/z$ 557.6, obs. $m/z$ 558.0.

Preparation of ethyl ester-conjugated gp91 ds-tat (egp91 ds-tat; Ac-Asp(OEt)-PEG4-PEG3-PEG4-gp91-NH$_2$) was carried out in the same way as for cgp91 ds-tat, except for replacement of Fmoc-Asp(OChol)-OH with Fmoc-Asp(OEt)-OH in the final coupling step: calcd. for C$_{148}$H$_{277}$N$_{56}$O$_{43}$S [M + 5H$^+$] $m/z$ 711.8, obs. $m/z$ 712.1; calcd. for C$_{148}$H$_{278}$N$_{56}$O$_{43}$S [M + 6H$^+$] $m/z$ 593.3, obs. $m/z$ 593.7; calcd. for C$_{148}$H$_{279}$N$_{56}$O$_{43}$S [M + 7H$^+$] $m/z$ 508.7, obs. $m/z$ 509.0.

Preparation of the 18 amino acid scrambled gp91 ds-tat (Sgp91 ds-tat; Ac-Asp (OChol)-PEG4-PEG3-PEG4-RKK-RRQRR-RCLRI-TRQSR-NH$_2$) peptide was carried out by manual SPPS as described above for unscrambled gp91 ds-tat. The resin-bound sgp91 ds-tat was then conjugated to cholestanol via a PEG linker using the same method described above for unscrambled cgp91 ds-tat. The crude peptide was purified in the same way to give cgp91 ds-tat: calcd. for C$_{162}$H$_{307}$N$_{54}$O$_{40}$S [M + 5H$^+$] $m/z$ 736.3, obs. $m/z$ 736.5; calcd. for C$_{162}$H$_{308}$N$_{54}$O$_{40}$S [M + 6H$^+$] $m/z$ 613.7, obs. $m/z$ 614.0; calcd. for C$_{162}$H$_{309}$N$_{54}$O$_{40}$S [M + 7H$^+$] $m/z$ 526.2, obs. $m/z$ 526.3

**Animal ethics statement**. The mouse experiments described in this manuscript were approved by the Animal Experimentation Ethics Committee of The

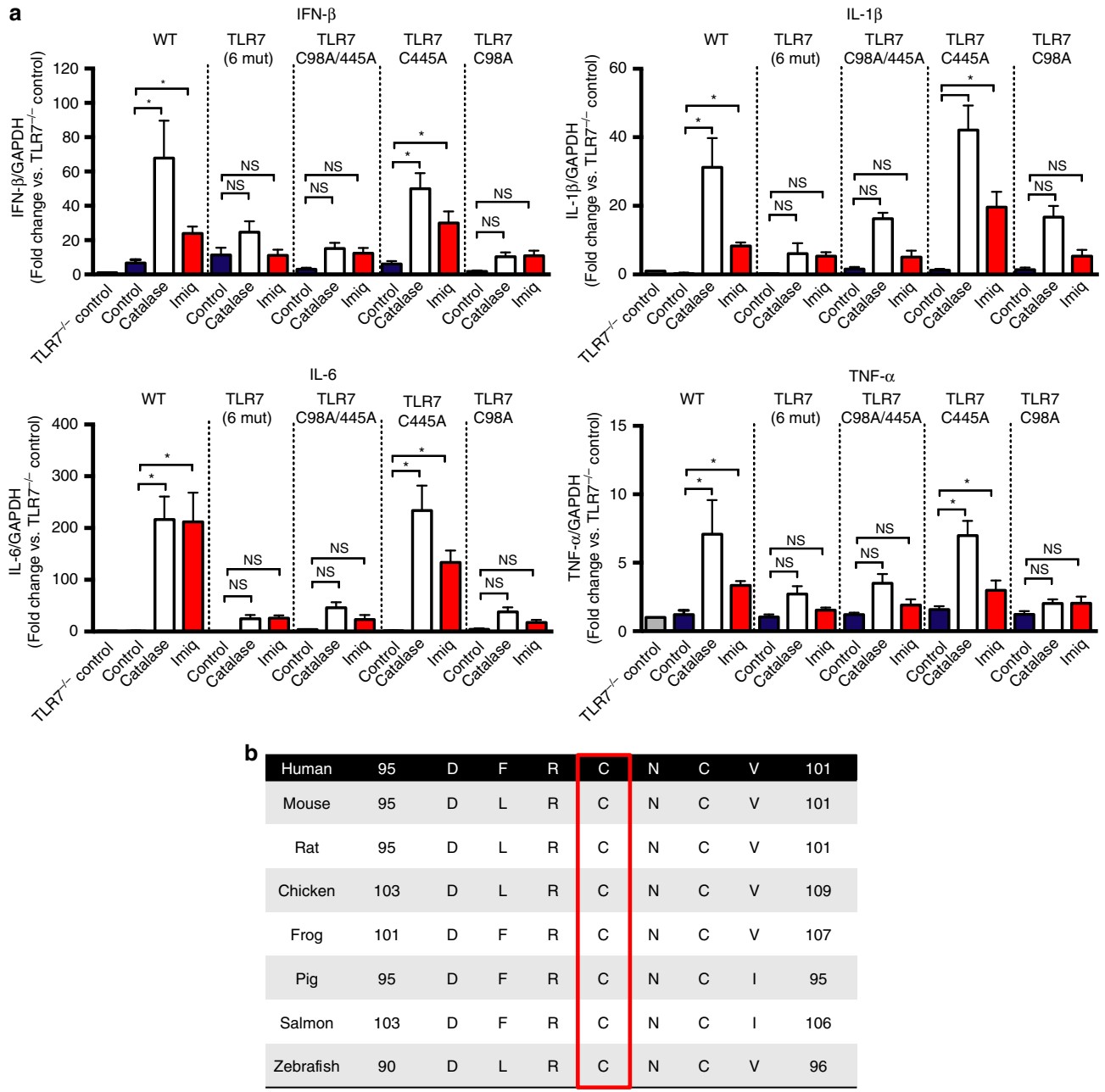

**Fig. 7** C98 on TLR7 regulates activity of the receptor and is a target for endosomal $H_2O_2$. **a** TLR7$^{-/-}$ BMDMs were transfected with empty vector, WT TLR7 or with either TLR7 with cysteines 98, 260, 263, 270, 273, and 445 mutated to alanine (TLR7 6 mut), TLR7 with cysteines 98 and 445 mutated to alanine (TLR7C98A/445 A) or with TLR7 with cysteines 445 (TLR7C445A) or 98 (TLR7C98A) mutated to alanine. After 48 h, cells were left untreated or treated for 1 h with either catalase (1000 U/ml) or imiquimod (Imiq, 10 μg/ml) and cytokine expression assessed 24 h later ($n = 6$). Responses are relative to GAPDH and then expressed as a fold-change above TLR7$^{-/-}$ controls. Data are represented as mean ± S.E.M. One-way ANOVA followed by Dunnett's post hoc test for multiple comparisons. Statistical significance was accepted when $P < 0.05$. *$P < 0.05$. (NS) Denotes not significant. **b** Multiple sequence alignment with CLUSTAL OMEGA showing across species conservation of Cys 98 on TLR7

University of Melbourne and Monash University and conducted in compliance with the guidelines of the National Health and Medical Research Council (NHMRC) of Australia on animal experimentation.

**In vivo infection with influenza A virus and drug treatments**. Aged matched (6–12 weeks) littermate male naïve WT control and NOX2$^{-/y}$ mice (also known as gp91phox$^{-/-}$ mice were originally generated in the laboratory of Prof Mary Dinauer[32]) were anaesthetized by penthrane inhalation and infected intranasally (i.n.) $1 \times 10^4$ or $1 \times 10^5$ plaque forming units (PFU) of Hk-x31 in a 35 μl volume, diluted in PBS. Mice were euthanised at Day 1, 3 or 7 following influenza infections. In some experiments, anaesthetized mice were treated via intranasal delivery with either dimethyl sulphoxide (DMSO, control; Sigma), unconjugated gp91dstat (0.02 mg/kg, 0.2 mg/kg), cholestanol conjugated-gp91dstat (0.02 mg/kg,

0.2 mg/kg) or cholestanol conjugated-scrambled gp91ds-TAT (0.02 mg/kg) one day prior to infection with Hk-x31 and everyday thereafter for 3 days. In additional experiments, anaesthetized mice were treated with imiquimod (50 μg per mouse, i. n.) or catalase (1000 U per mouse, i.n.) and then euthanised for analysis at Day 1.

**Airways inflammation and differential cell counting**. Mice were killed by an intraperitoneal (i.p.) injection of ketamine/xylazene (100 mg/kg) mixture. An incision was made from the lower jaw to the top of the rib cage, where the salivary glands were separated to expose the surface of the trachea. The layer of smooth muscle on the trachea was removed, allowing a small incision to be made near the top of the trachea. A sheathed 21-Gauge needle was inserted to the lumen and 300–400 μl of PBS was lavaged repeatedly (four times). The total number cells in the BALF were stained with 0.4% trypan blue solution (Thermofisher Scientific,

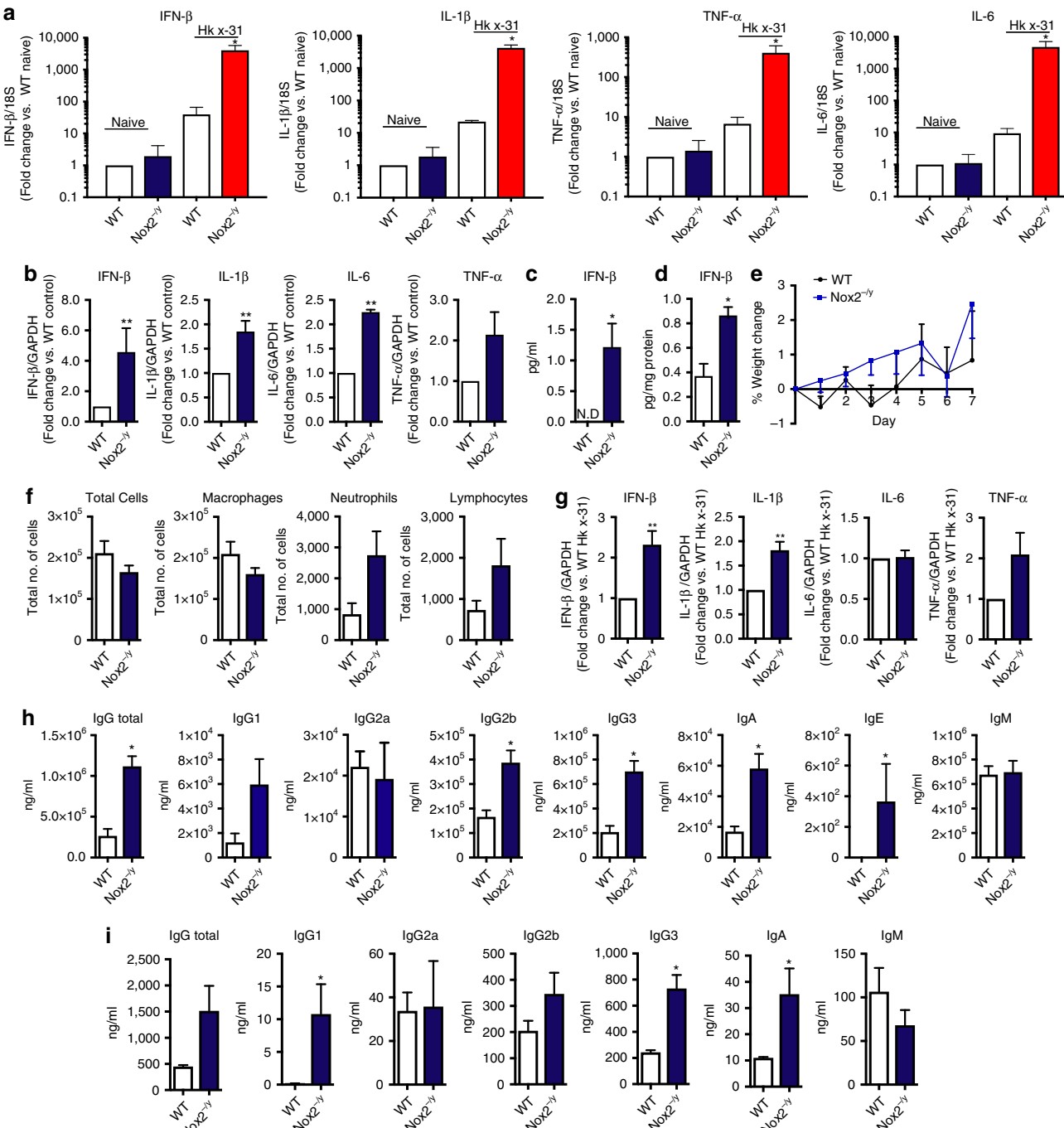

**Fig. 8** Inhibition of NOX2 oxidase increases expression of Type I IFN and IL-1β, and antibody production to influenza A virus infection. **a** Alveolar macrophages from WT and Nox2$^{-/y}$ mice were either left untreated (naïve) or infected with HKx31 influenza A virus (MOI of 10) for analysis of IFN-β, IL-1β, TNF-α, and IL-6 mRNA expression by QPCR after 24 h ($n = 8$). **b**, **c** WT and Nox2$^{-/y}$ mice were infected with live HKx31 influenza A virus ($1 \times 10^5$ PFU per mouse) and **b** cytokine mRNA expression and IFN-β protein expression in **c** BALF or **d** serum were assessed 3 days later ($n = 5$). **e–i** WT and Nox2$^{-/y}$ mice were infected with inactivated HKx31 influenza A virus (equivalent to $1 \times 10^4$ PFU per mouse) for measurements at day 7 of: **e** body weight; **f** airway inflammation and differential cell counts (i.e., macrophages, neutrophils, and lymphocytes); **g** cytokine expression in whole lung (responses are shown as fold change relative to GAPDH) and **h** serum and **i** BALF antibody levels ($n = 6$). Data are shown as mean ± SE. **a** Kruskal–Wallis test with Dunn's post hoc for multiple comparisons. **b–i** Unpaired t-test; statistical significance taken when the $P < 0.05$. *$P < 0.05$. **$P < 0.01$

USA) and viable cells were evaluated using the Countess automated cell counter (Invitrogen, Cat # C10227). Differential cell analysis was prepared from BALF ($5 \times 10^4$ cells) that were centrifuged at $3 \times g$ for 5 min on the Cytospin 3 (Shandon, UK). Following this, slides were fixed in 100% propanol for 1 min and allowed to dry overnight. Finally, samples were stained with Rapid I Aqueous Red StainTM (AMBER Scientific, Australia) and Rapid II Blue StainTM (AMBER Scientific, Australia) for 10 min, then submerged in 70% ethanol and absolute ethanol twice

before being placed into xylene for 5 min (two times). Samples were then mounted in DPX mounting medium (Labchem, NSW, Australia) and coverslips were firmly placed on top. 500 cells per sample from random fields were differentiated into macrophages, neutrophils, eosinophils and lymphocytes by standard morphological criteria. Data are represented as total cell numbers that was calculated by the respective cell type multiplied by the total live cell numbers and as a percentage of the cell population.

**Cell culture and primary cell isolation**. Human AMs were obtained from subjects undergoing a bronchoscopy at Monash Medical Centre, Monash University to investigate underlying lung disease with approval from the ethics committee of Southern Health/Monash Medical Centre. Written consent was obtained from all subjects. The bronchoscope was wedged in the right middle lobe and 25–50 ml of saline was washed into the airway then aspirated. Cells were washed twice with PBS before being suspended in culture medium (Roswell Park Memorial Institute (RPMI, Life Technologies, Cat # 21870-076) with 10% FCS with 100 U/ml penicillin and 100 µg/ml streptomycin) for ~24 h before use.

AMs were isolated by lung lavage from age-matched (6–12 weeks) male C57Bl/6 J (WT), NOX2$^{-/y}$, NOX4$^{-/-}$ (provided by Dr Hitesh Peshavariya, Centre for Eye Research, The University of Melbourne, Australia), TLR7$^{-/-}$ (provided by Prof Phil Hansbro, School of Biomedical Sciences and Pharmacy, Faculty of Health and Medicine, The University of Newcastle, and Hunter Medical research Institute, New South Wales, Australia), TLR9$^{-/-}$ (provided by Prof. Karlheinz Peter (Baker IDI Heart & Diabetes Institute, Melbourne, Victoria, Australia)) or SOD3$^{-/-}$ mice (provided by A/Prof Steven Bozinovski, School of Health and Biomedical Sciences, RMIT University). A thin shallow midline incision from the lower jaw to the top of

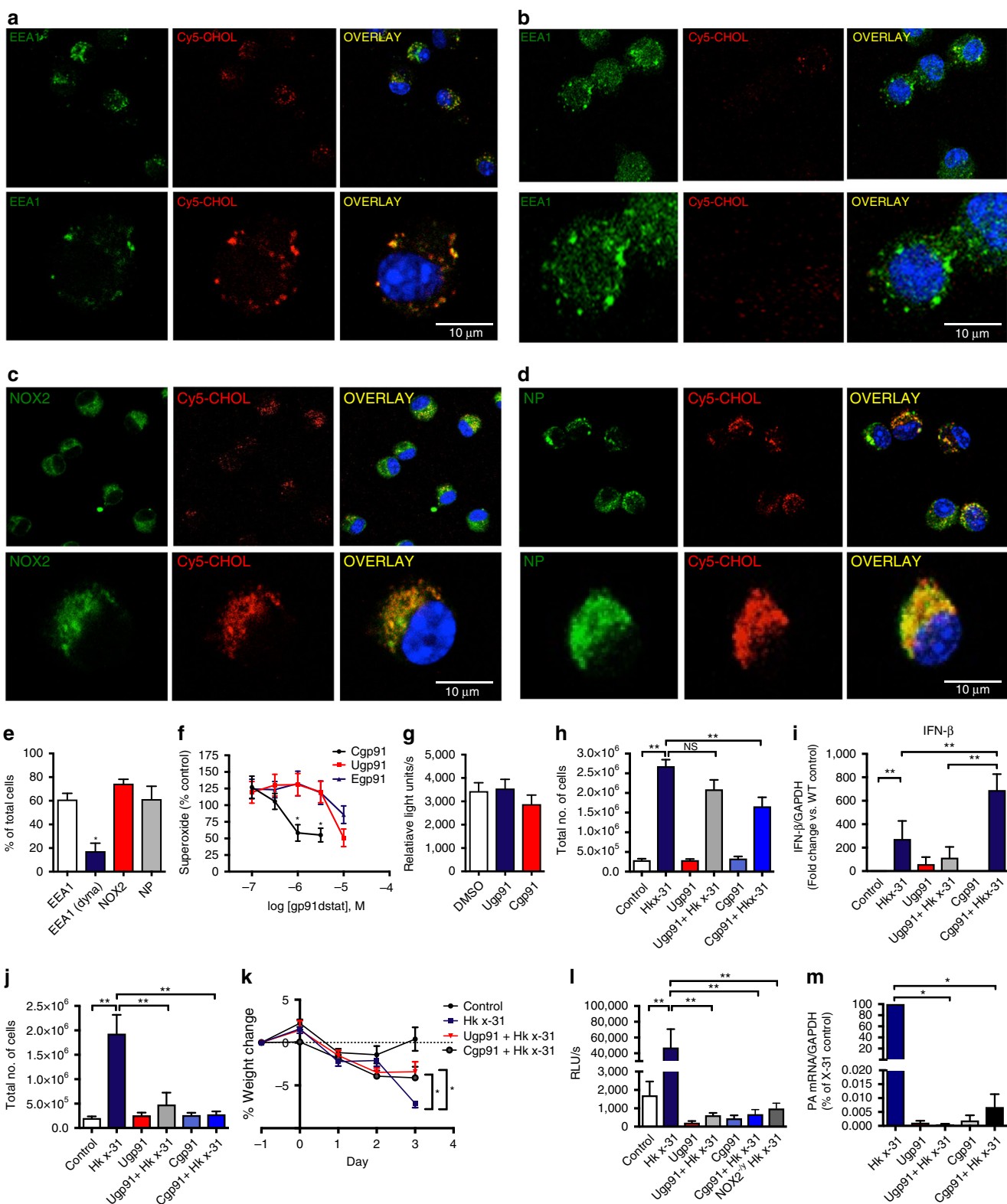

the rib cage was made and the larynx was separated to expose the top of the trachea. The layer of smooth muscle covering the trachea was removed, a small incision made and a sheathed 21-Gauge needle was inserted into the lumen. The lungs were repeatedly (three times) lavaged with 300–400 μl of PBS. Cell suspensions were spun down by centrifugation ($200 \times g$ at 4 °C for 5 min). Supernatant was removed, then cells were re-suspended in 1 ml of sterile PBS and counted using the Countess automated cell counter (Invitrogen, Cat # C10227). Cells were then seeded into 24-well plates ($1 \times 10^5$ cells per well) for immunocytochemistry and fluorescence microscopy, as stated below.

The immortalized cell line RAW 264.7 cells (ATCC TIB-71) and immortalized BMDMs (courtesy of Dr Ashley Mansell of the Hudson Institute of Medical Research Monash University and Prof Eicke Latz of the Institute of Innate Immunity, University of Bonn, Germany) were maintained in Dulbecco's Modified Eagle's Medium (DMEM: Sigma) supplemented with L-glutamine, glucose (4500 mg/l), sodium pyruvate (110 mg/l), and fetal bovine serum (FBS; 10%). The TLR2$^{-/-}$, TLR3$^{-/-}$, TLR4$^{-/-}$, TLR7$^{-/-}$, TRIF$^{-/-}$, RIG-I$^{-/-}$, MyD88$^{-/-}$, NLRP3$^{-/-}$, and UNC93B1$^{-/-}$ immortalized BMDMs (provided by Dr. Ashley Mansell, Dr Michael Gantier and Prof. Eicke Latz) were maintained in RPMI medium supplemented with glucose (4500 mg/l), non essential amino acids, sodium pyruvate, streptomycin and FBS (10%) and DMEM (20%) (containing all supplements, as stated above). All cells were kept at 37 °C with a humidified mixture of 5% $CO_2$ and 95% air. The medium was changed two to three times a week, cells were sub-cultured by scraping when ~80–90% confluent, and counted using the Countess automated cell counter.

**Confocal fluorescence microscopy.** Cells were seeded onto glass cover slips in 24-well plates, and allowed to adhere for 24 h in DMEM. Cells were then incubated in the absence or presence of HKx31 influenza A virus (MOI 0.1, 1 or 10) in serum-free medium at varying time points (5, 15, 30 min, and 1 h). In some cases, cells were pretreated for 30 min prior to infection with Dynasore (100 μM) or the vehicle for Dynasore, DMSO (0.1%). Next, the cells were washed with PBS (0.01 M) and fixed with 4% paraformaldehyde (PFA) for 15 min. Cells were treated for 10 min with PBS-containing Triton X-100 (0.25%) and then washed three times over 15 min with PBS. The samples were then incubated with 10% goat serum-containing PBS for 2 h and/or mouse on mouse IgG blocking reagent (Cat # MKB-2213, Vector Laboratories). This was followed by the addition of a primary antibody for nucleoprotein (1:1000) to localize influenza A virus, purified mouse anti-NOX2 (1:500) to localize NOX2, rabbit anti-TLR7 (1:1000) to localize TLR7, or mouse anti-early endosome antigen 1 (EEA1: 1000) to localize early endosomes for 24 h at 4 °C. In some experiments, combinations of antibodies were used at the indicated concentrations to determine protein co-localization. Cells were washed three times over 30 min with PBS (0.01 M). Following the washes, a secondary antibody goat anti-rabbit alexa 594 (1:1000), goat anti rabbit red 647 (1:500, 1:1000) and/or biotinylated anti-mouse IgG was added to appropriate wells in the dark for 2 h. Finally, the cells were washed three times over 30 min with PBS (0.01 M); and where appropriate (mouse primary and secondary anti Fluroscein Avidin DCS was applied for 5 min). Cover slips were mounted onto microscope slides with 10–20 μl of diamidino-2-phenylindole (DAPI) for 3 min. Slides were viewed and photographed on a Nikon upright inverted confocal fluorescence microscope (Nikon D-eclipse C1). All immunohistochemistry was assessed by two observers blinded as to the treatment groups throughout the analysis process and all of the appropriate controls were performed, in that all combinations of primary and secondary antibodies were used to ensure no cross reactivity occurred.

We verified the specificity of both the TLR7 and NOX2 antibodies by examining the degree of staining in AMs taken from WT, TLR7$^{-/-}$ and NOX2$^{-/-}$ mice, respectively. There was no staining for TLR7 in the TLR7$^{-/-}$ macrophages (Fig. 2c). Similarly with the NOX2 antibody we observed no staining in AMs of NOX2$^{-/y}$ mice compared to the WT cells (Supplementary Fig. 1d). Further evidence for the specificity of this NOX2 antibody can be found in Judkins et al.[33].

**Endosomal ROS production.** Human AMs; WT, TLR7$^{-/-}$, TLR2$^{-/-}$, TLR3$^{-/-}$, TLR4$^{-/-}$, TRIF$^{-/-}$, MyD88$^{-/-}$, RIG-I$^{-/-}$ and NLRP3$^{-/-}$ BMDMs; mouse primary WT, NOX2$^{-/y}$, NOX4$^{-/-}$, TLR7$^{-/-}$, TLR9$^{-/-}$ or SOD3$^{-/-}$ AMs and RAW264.7 cells were seeded ($1 \times 10^5$ cells per well) onto glass coverslips in 24-well plates allowing the cells to adhere for 24 h in DMEM or RPMI medium before being pretreated with OxyBURST Green H2HFF (100 μM) and/or LysoTracker Deep Red (50 nM) for 5 min. This was followed by incubation with PBS (control group; 0.01 M), imiquimod (10 μg/ml), single stranded RNA (ssRNA; 100 μM), or infected with either H3N2 influenza viruses (A/New York/55/2004, A/Brisbane/9/2007), seasonal H1N1 influenza A viruses (A/Brazil/11/1978, A/New Caledonia/20/1999, A/Solomon Islands/3/2006), A(H1N1)pdm influenza A viruses (A/California/7/2009, A/Auckland/1/2009), or with re-assortant vaccine strains HKx31 (MOI 0.1–10) or BJx109 (MOI 10) in serum-free medium at varying time points (5, 15, 30 min, and 1 h). Other wells were infected with dengue virus (MOI 10), Sendai virus (40 HAU/ml), human parainfluenza virus (MOI 10), human metapneumovirus (MOI 10), rhinovirus (MOI 10), respiratory syncytial virus (MOI 10), HIV (MOI 10), Newcastle disease virus (MOI 10), mumps virus (MOI 10), rhesus or UK rotaviruses (each at MOI 10) or herpes simplex virus-2 (MOI 10) under similar conditions. In some cases, cells were pretreated with SOD (300 U/ml), apocynin (300 μM), gp91dstat (50 μM) or bafilomycin A (100 nM), for 30 min prior to infection. Next, the cells were washed with PBS (0.01 M) and fixed with 4% PFA for 15 min. After fixation, cells were then washed three times with PBS over 30 min. Cover slips were then mounted onto microscope slides with 10–20 μl of DAPI for 3 min, then analyzed and photographed on an Nikon upright confocal fluorescence microscope (Nikon D-eclipse C1).

**NOX2 oxidase assembly.** To measure NOX2 oxidase activity we assessed p47phox and NOX2 assembly using confocal fluorescence microscopy. Control and HKx31 virus-infected WT and TLR7$^{-/-}$ AMs were processed as indicated above under "confocal fluorescence microscopy". In additional experiments, WT cells were treated with Dynasore (100 μM) or bafilomycin A (100 nM) for 30 min prior to virus infection. After exposing samples with 10% goat serum-containing PBS for 2 h, the rabbit anti-p47phox antibody (1:1000) and the mouse anti-NOX2 antibody (1:500) were added followed by addition of appropriate secondary antibodies, as specified above.

**L-O12-enhanced chemiluminescence.** ROS production was quantified using L-O12-enhanced chemiluminescence. RAW264.7 cells and primary mouse AMs were seeded into a 96-well OptiView plate ($5 \times 10^4$ cells per well). RAW264.7 cells were either treated with DMSO (control, appropriate concentration), unconjugated gp91dstat (100 nM–30 μM), cholestanol-conjugated gp91dstat (100 nM–30 μM) or ethyl-conjugated gp91dstat (100 nM–30 μM) for 1 h. BALF was collected from mice treated with DMSO (control), unconjugated gp91dstat (0.02 mg/kg, 0.2 mg/kg), cholestanol conjugated-gp91dstat (0.02 mg/kg, 0.2 mg/kg) and/or infected with Hk-x31 influenza A virus ($1 \times 10^5$ PFUs). Cells were then washed of media with 37 °C Krebs-HEPES buffer, then exposed to a Krebs-HEPES buffer containing L-O12 ($10^{-4}$ mol/l) in the absence (i.e., basal ROS production) or presence (stimulated ROS production) of the PKC and NADPH oxidase activator phorbol dibutyrate (PDB; $10^{-6}$ mol/l). The same treatments were performed in blank wells (i.e., with no cells), which served as controls for background luminescence. All treatment groups were performed in triplicates. Photon emission [relative light units (RLU)/s] was detected using the Chameleon luminescence detector (Hidex, model 425105, Finland) and recorded from each well for 1 s over 60 cycles. Individual data points for each group were derived from the average values of the three replicates minus the respective blank controls. Data are represented as a % of the control in the dose-response curves or as raw values (ex vivo experiments).

**Fig. 9** Endosome targeted delivery of a NOX2 oxidase inhibitor protects mice following influenza A virus infection in vivo. **a–e** Alveolar macrophages from WT mice were treated with the Cy5 cholestanol-PEG linker fluorophore (Cy5-chol; 100 nM) for 30 min and infected with HKx31 influenza A virus (MOI of 10). Cells were then counter labeled with antibodies to either: **a** and **b** EEA1, **c** NOX2 or **d** influenza A nucleoprotein (NP). All cells were then stained with 4′,6′-diamidino-2-phenylindole (DAPI) and imaged with confocal microscopy. **b** Cells were pretreated with dynasore (100 μM) for 30 min prior to exposure to Cy5-cholestanol. **e** Quantification of data from (**a–d**, $n = 5$). **f** RAW 264.7 macrophages were either untreated or treated with various concentrations of cholestanol-conjugated gp91ds-TAT (Cgp91), ethyl conjugated gp91ds-TAT (Egp91) or unconjugated gp91ds-TAT (Ugp91) for 30 min prior to quantifying ROS production by L-O12 (100 μM)-enhanced chemiluminescence ($n = 7$). **g** Superoxide production via the xanthine/xanthine oxidase cell-free assay in the absence or presence of Ugp91ds-TAT, (1 μM) or Cgp91ds-TAT (1 μM) ($n = 6$). **h, i** Ugp91ds-TAT (0.02 mg/kg/day) or Cgp91ds-TAT (0.02 mg/kg/day) were delivered intranasally to WT mice once daily for 4 days. At 24 h after the first dose of inhibitor, mice were either treated with saline or infected with HKx31 influenza A virus ($1 \times 10^5$ PFU per mouse). Mice were culled at day 3 post-infection and **h** airway inflammation was assessed by BALF cell counts and **i** lung IFN-β mRNA was determined by QPCR ($n = 7$). (NS) denotes not significant. **j–m** Mice were subjected to the NOX2 inhibitor treatment regime and virus infection protocol as in **h** except NOX2 inhibitors were delivered at a dose of 0.2 mg/kg/day ($n = 7$). Analysis of **j** airway inflammation by BALF counts, **k** body weight (% weight change from the value measured at Day -1), **l** ROS production by BALF inflammatory cells with L-O12 enhanced chemiluminescence and **m** viral mRNA by QPCR. Data are represented as mean ± S.E.M. **e** Unpaired t-test; statistical significance taken when the $P < 0.05$. **f, g, h, j, k, l** One-way ANOVA followed by Dunnett's post hoc test for multiple comparisons. **i** and **m** Kruskal–Wallis test with Dunn's post hoc for multiple comparisons. Statistical significance was accepted when $P < 0.05$. *$P < 0.05$; **$P < 0.01$. Scale bars: 10 μm

To test whether the unconjugated or cholestanol conjugated gp91dstat exhibited ROS scavenging properties, the xanthine oxidase cell free assay was used. Briefly, Krebs-HEPES buffer containing L-012 (100 µM) was added into a 96-well Optiview plate. Following this, 0.1% DMSO, unconjugated gp91dstat (Ugp91ds-TAT, 1 µM) or cholestanol-conjugated gp91ds-TAT (1 µM) were added in combination with Xanthine (100 µM). Immediately after xanthine oxidase (0.03 U/ml) was added, photon emission [relative light units (RLU)/s] was detected using the Chameleon luminescence detector (Hidex, model 425105, Finland) and recorded from each well for 1 s over 60 cycles. Individual data points for each group were derived from the average values of the three replicates minus the respective blank controls. Data are represented as raw values.

**Site directed mutagenesis, sequencing and transfections**. HA-TLR7 cDNA was purchased from Sino Biological (mouse TLR7; Cat # MG50962-NY with Gene Bank Ref Seq number NM_133211.3). Mutation of the key cysteine residues in TLR7 (Cys260, Cys263, Cys270, Cys273, Cys98, and Cys445) to alanine was performed using the QuikChange Multi Site-Directed Mutagenesis kit (Cat # 200514, Agilent Technologies). Sequences of WT and mutant HA-TLR7 were confirmed by the Australian Genome Research Facility. Cells were transfected using linear polyethyleneimine (PEI)[24].

**High-content ratiometric FRET imaging**. Cells were plated and transfected in suspension with 200 ng per well FRET biosensor DNA using PEI, in black, optically clear 96-well plates for 48 hr. Prior to the experiment, cells were partially serum-starved overnight in 0.5% FBS media. Fluorescence imaging was performed using a high-content GE Healthcare INCell 2000 Analyzer with a Nikon Plan Fluor ELWD 40×(NA 0.6) objective and FRET module as described[22]. For CFP/YFP (CKAR) emission ratio analysis, cells were sequentially excited using a CFP filter (430/24) with emission measured using YFP (535/30) and CFP (470/24) filters, and a polychroic optimized for the CFP/YFP filter pair (Quad3). For GFP/RFP (EKAR) emission ratio analysis, cells were sequentially excited using a FITC filter (490/20) with emission measured using dsRed (605/52) and FITC (525/36) filters, and a polychroic optimized for the FITC/dsRed filter pair (Quad4). Cells were imaged every 100 s for 20 min (image capture of two fields of view in 12 wells per 100 s). Data were analyzed using in-house scripts written for the FIJI distribution of Image J[34], as described[24].

**Quantification of mRNA by QPCR**. Cells were treated with imiquimod (10 µg/ml), poly I:C (100 ng/ml), CpG (10 µg/ml), ssRNA (500 µg/ml) or catalase (1000 U/ml) for 24 h. Where indicated, cells were pre-treated with apocynin (300 µM), SOD (300 U/ml) or bafilomycin A (100 nM) for 30 min. RNA was extracted from the lung tissue of mice that were treated with either DMSO (control), unconjugated gp91dstat (0.02 mg/kg, 0.2 mg/kg), cholestanol conjugated-gp91dstat (0.02 mg/kg, 0.2 mg/kg), scrambled cholestanol conjugated gp91dstat (0.02 mg/kg) and/or infected with Hk-x31 influenza A virus ($1 \times 10^5$ PFUs) 3 days post infection for the assessment of viral mRNA and cytokine expression. The right lung lobe was placed in Eppendorf tubes containing a mixture of Buffer RLT (Qiagen, USA) and β-mercaptoethanol (Sigma; 1%), which was minced into small pieces using curved scissors. Following this, lung samples were homogenized using the ultrasound homogenizer (Hielscher Ultrasonics GmBH, Teltow, Germany) and centrifuged at $18407 \times g$ for 5 min. A 1:1 ratio of lysate was mixed with 70% RNase free ethanol transferred to RNeasy spin columns (RNeasy Minikit; Cat # 74104, Qiagen). Samples were spun at 10,000 rpm for 15 s and then washed with Buffer RW1. After discarding the flow-through, 5 µl of DNase I (Cat # 79254, Qiagen) was mixed with 35 µl of Buffer RDD was pipetted directly onto the membrane of the spin column and incubated at room temperature for 15 min. Buffer RPE was added and centrifuged for $9391 \times g$ for 15 s. After discarding the flow-through, Buffer RPE was re-added and spun for $9391 \times g$ for 2 min. An additional spin at $18407 \times g$ for 1 min was done to remove residual flow-through from the spin column. Finally, RNase free water was added and centrifuged to elute the RNA into an Eppendorf tube. RNA samples were measured using the Nanodrop 1000 Spectrophotometer (Thermo Scientific, USA). cDNA synthesis was performed using the High-Capacity cDNA Reverse Transcription Kit (Cat # 4368814, Applied Biosystems, Foster City, CA, USA) using 1.0–2.0 µg total RNA. RNA was added to a mixture of reagents in the High-Capacity cDNA RT kit to make a final volume of 20 µl. This was transcribed using the BioRad MycyclerTM thermal cycler (BioRad, USA) at the following settings: 25 °C for 10 min, 37 °C for 120 min, 85 °C for 5 min and 4 °C at rest. Samples were stored at −20 °C prior to use. Quantitative polymerase chain reaction was carried out using the TaqMan Universal PCR Master Mix (Cat # 4304437, Applied Biosystems, Foster City, CA, USA) or SYBR Green PCR Master Mix (Cat # 4367659, Applied Biosystems, Foster City, CA, USA) and analyzed on ABI Step One TM and StepOnePlusTM Real-time PCR Systems (Perkin-Elmer Applied Biosystems, Foster City, CA, USA). The PCR primers for TNF-α, IL-1β, IFN-β and IL-6 were included in the Assayon- Demand Gene Expression Assay Mix (see Supplementary Table 2, Applied Biosystems, Foster City, CA, USA). Additionally, a custom designed forward and reverse primer of the segment 3 polymerase (PA) of influenza virus was used to measure viral titers (Supplementary Table 2). The PCR program run settings: 50 °C for 2 min, followed by 95 °C for 1 h, then 95 °C for 15 s + 60 °C for 60 s + plate read (40 cycles). Quantitative values 129 were obtained from the threshold cycle (Ct) number. Target gene expression level was normalized against 18 s or GAPDH mRNA expression for each sample and data was expressed relative to the control.

**ELISA and multiplex immunoassay**. Protein levels of IFN-β (VeriKine HM mouse IFN β Serum ELISA kit; Cat # 42410-1, PBL Assay Science), IL-1β (Quantikine ELISA Mouse IL-1β/IL-IF2; Cat # MLB00C, R&D Systems), TNF-α (Quantikine ELISA Mouse TNF α, Cat # MTA00B, R&D Systems), and IL-6 (Quantikine ELISA Mouse IL-6, Cat # M6000B, R&D Systems) secreted into the BALF of HKx31-infected ($1 \times 10^4$ PFUs) WT and NOX2$^{-/y}$ mice were measured using ELISAs and performed using commercially available kits according to the manufacturer's instructions. The cytokine titers in samples were determined by plotting the optical densities using a 4-parameter fit for the standard curve.

**Antibody determination**. Serum and BALF levels of various antibody isotypes (IgA, IgE, IgG1, Ig2a, IgG2b, IgG3, IgM, and total IgG) were quantified in HKx31-infected ($1 \times 10^4$ PFUs) WT and NOX2$^{-/y}$ mice using the ProcartaPlex Multiplex Immunoassay (Mouse Isotyping 7plex, Cat # EPX070-2815-901, eBioscience) according to the manufacturer's instructions. Briefly, antibody-conjugated magnetic beads were added into each well of a 96-well plate. Antibody standards were serially diluted (1:4) in universal assay buffer to construct a 7-point standard curve. Serum and BALF samples (diluted 1:20,000 in universal assay buffer) and/or standards were added to appropriate wells of the 96-well plate containing the antibody-conjugated magnetic beads. Following this, a detection antibody mix was added to each well and the plate was incubated for 30 min at room temperature on a microplate shaker (500 rpm) in the dark. After washing, a reading buffer was added to all wells. The plate was read by a Magpix multiplex reader (Luminex, USA) with xPONENT software (Luminex, USA). Procartaplex Analyst 1.0 software (eBioscience, USA) was used to interpolate serum and BALF antibody concentrations in each sample from the standard curve.

**Statistical analysis and image analysis**. In order to quantify the fluorescence microscopy data, images acquired from confocal systems were analyzed in Image J. Approximately 100–150 cells per treatment group from at least three independent experiments were analyzed unless otherwise stated in the figure legend to calculate the fluorescence in each cell, which was then averaged and expressed as a percentage of the area fluorescence. All statistical tests were performed using GraphPad Prism (GraphPad Software Version 6.0, San Diego CA, USA). $P < 0.05$ was taken to indicate significance. For isolated cell culture work, $n$ is representative of a separate experiment where cells were used from a different passage.

**Chemicals**. Imiquimod (Cat # tlrl-imq, Invivogen), HMW poly I:C (Cat # tlrl-pic, Invivogen), and CpG ODN (Cat # tlrl-1668, Invivogen) were dissolved in endotoxin-free water and prepared as stock solutions of 5–10 mg/ml in aliquots of 30 and 100 µl and stored at −20 °C. ssRNA (Cat # tlrl-lrna40, Invivogen) was dissolved in endotoxin-free water and prepared as a stock solution of 5 mM in aliquots of 50 µl and stored at −20 °C. Dynasore (Cat # D7693, Sigma) (freshly prepared on the day) was dissolved in DMSO (100%) and prepared as 10 mM stock solutions. FBS (Cat # 12003 C, Sigma) was stored in 50 ml aliquots at −20 °C. Penicillin–streptomycin solution (Cat # P4333, Sigma) was stored at −20 °C. SOD (Cat # S2515, Sigma) was dissolved in distilled water and prepared as stock solutions (10 µl) of 30,000 units per ml and stored at −20 °C. OxyBURST Green H2HFF bovine serum albumin (BSA) (Cat # 1329, Molecular probes, Life Technologies) and LysoTracker Deep Red (Cat # L12492, Molecular probes, Life Technologies) were generated immediately before use by dissolving in PBS. Bafilomycin A (from *Streptomyces*, Cat # B1793, Sigma) was prepared as a stock solution of 100 µM in aliquots of 10 µl and stored at −20 °C. Apocynin (4′-Hydroxy-3′-methoxy acetophenone, Cat # A10809, Sigma) made freshly on the day of use and gp91dstat (Cat # AS-63818, Anaspec) were prepared as stock solutions of 100 and 50 mM respectively, in 100% DMSO. Phorbol dibutyrate (Cat # P1239, Sigma) was dissolved in 100% DMSO as 10 mM stocks and made fresh on the day of use. Catalase (Cat # C1345, Sigma) was prepared as stock solutions of $10^6$ U/ml in distilled water and stored at −20 °C. MitoSOX (Cat # M3600850, Molecular Probes, Life Technologies) was prepared at 5 mM by dissolving the contents (50 µg) of one vial of MitoSOX mitochondrial superoxide indicator (Component A) in 13 µl of DMSO. Xanthine oxidase (Cat # X1875, Sigma) was prepared fresh on the day by dissolving in distilled water to 30 U/ml and xanthine (Cat # X0626, Sigma) was prepared as a stock of 100 mM in 0.1 M NaOH. ML171 (Cat # 492002, Calbiochem).

Antibodies for influenza nucleoprotein (mAb to Influenza A Virus Nucleoprotein [AA5H]; Cat # 120-20343, AbCAM), early endosome antigen 1 (Cat # 120-02900, AbCAM), mouse anti-gp91phox (Cat # 611415, BD Transduction Laboratories, Purified Mouse Anti gp91[phox] Clone 53/gp91[phox] (RUO), rabbit anti-TLR7 (Cat # NBP2-24906, Novus Biologicals), rabbit anti-p47phox antibody (Cat # sc14015, Santa Cruz), FITC goat anti-mouse IgG (Cat # A-11029, Invitrogen), goat anti-rabbit alexa 594 (Cat # A-11037, Invitrogen), goat anti-rabbit far red 647 (Cat # A-21244, Invitrogen), and DAPI (Cat # H-1200, Vector Laboratories) were stored at −20 °C.

**Data availability**. The data that support the findings of this study are available from the corresponding author upon request.

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

## Acknowledgements

This work was supported by the Australian Research Council (ARC) Future Fellowship Scheme for S.S. (I.D. FT120100876) and S.B. (I.D. FT130100654); The National Health and Medical Research Council of Australia (NHMRC) RD Wright Fellowship scheme for M.L.H. (I.D. 1061687); The NHMRC Senior Research Fellowship Scheme for G.R.D. (I.D. 1006017) and C.G.S. (I.D. 1079467), The NHMRC practitioner fellowship for S.R.L. (I.D. 1042654); The NHMRC Early Career Fellowship for M.R.S. (I.D. 1072000); The NHMRC project grant schemes (Project I.D. 1122506, 1128276, 1027112, 1041795, 1052979); the ARC Centre of Excellence in Bio-Nano Science and Technology (Project CE140100036) and The Australian Postgraduate Award for E.E.T. The authors acknowledge the Monash Micro-Imaging facility (provision of instrumentation and training). The authors also wish to thank the following people for providing viruses including Associate Prof. Elizabeth McGraw (Monash University); A/Prof. Barbara Coulson (The Peter Doherty Institute for Infection and Immunity, The University of Melbourne); Dr Ashley Mansell (Hudson Institute of Medical Research, Monash University), Dr Niamh Mangan (Hudson Institute of Medical Research, Monash University), A/Prof David Tscharke (Australia National University), Prof Sharon Lewin (The Peter Doherty Institute for Infection and Immunity, The University of Melbourne) and A/Prof John Stambas (CSIRO, Deakin University, Geelong, Australia). We thank Prof Karlheinz Peter (Baker IDI Heart and Diabetes Institute, Melbourne, Australia) for providing TLR9 −/− mice and Prof Philip Hansbro (Hunter Medical School, University of Newcastle, Australia) for TLR7−/− mice. Also the authors wish to thank Prof Arthur Christopoulos (Drug Discovery Biology, Monash institute of Pharmaceutical Sciences, Monash University, Australia) for providing feedback on the manuscript and Ms Felicia Liong (School of Health and Biomedical Sciences, RMIT University) for proof reading the manuscript.

## Author contributions

E.E.T., R.L., M.L.H., C.C., B.R.S.B., R.v.d.S., T.Q., S.B., R.V., S.S.: Performed experiments. E.E.T. and S.S: Wrote the manuscript. P.C.R., P.T.K., S.R.L., G.R.D., C.G.S., B.R.S.B., M.R.S., C.J.H.P., S.B., R.V., L.A.J.O., D.A.B., J.J.O. and S.S.: Provided intellectual input and edited the manuscript. P.C.R.: Provided influenza viruses and P.T.K.: Provided human alveolar macrophages. S.S.: Supervised and managed the overall study.

## Additional information

**Competing interests:** The authors declare no competing financial interests.

