## [Peer Review File · Nature Communications]

Reviewers' comments:

Reviewer #1 (Remarks to the Author):

This is an interesting MS suggesting that respiratory viruses activate the phagocyte NADPH oxidase in an endosomal compartment and that this activation limits the anti-viral response. I have the following comments and questions:

- The legend to figure 1 is quite confusing, but as far as I understand, the authors show that bafilomycin inhibits the oxyburst signal and suggest that inhibition of endosomal acidification inhibits endosomal ROS generation by NOX2. Why would that be the case? how can the authors exclude that this is an artefact of the measurement technique.

- There are reasonable doubts about the quality and the specificity of the mouse NOX2 antibody used by the authors. Thus, the authors should show a very careful documentation of the quality and specificity of the antibody

- The authors show that NOX2 activity reduces TLR-7 dependent signaling and hence there is an increased inflammatory response in NOX2-deficient phagocytes. The authors contribute this phenomenon to oxidation of cysteine residues in TLR-7. However the authors fail to acknowledge that the phenomenon of hyperinflammation in the absence of NOX2 is not at all specific for TLR-7. For example, very similar phenomena have been described in detail for fungal products and the activation of dectin-1 (e.g. J Pathol. 2008 Mar;214(4):434-44). Thus, the phenomenon of hyperinflammation is not TLR-7 specific and the proposed mechanism appears too simplistic.

- I am also not fully convinced about the endosomal localization of the NOX2-dependent ROS generation. Indeed, the fact that the oxyburst measurement detect a signal does not prove that the ROS are generated in the endosomes. Indeed, H₂O₂ diffuses well and permeates membranes; therefore its site of generation and site of detection might be distinct.

- I see a similar problem with the endosomal NOX inhibitor Egp91 ds-tat. Indeed, the suggested site of action of the peptide inhibitor is on the cytosolic part of the NOX2 complex. Thus, the inhibitor needs to cross the endosomal (or plasma) membrane to efficiently inhibit. However once the compound has crossed the membrane, how can the authors be sure that it still acts locally around the endosome and does not diffuse towards other sites of NOX2 localization?

Reviewer #2 (Remarks to the Author):

The manuscript by To et al. is very exciting and describes a role of endosomal NOX2 oxidase (and ROS production) in negatively modulating viral RNA mediated TLR7 activation. Furthermore targeting endosomal ROS production may serve as a new antiviral therapeutic strategy.

The manuscript is quite dense and overloaded, but very interesting.

The experiments concerning ROS-induction via NOX2 oxidase activation by influenza (Fig. 1) and colocalization of TLR7 with various RNA viruses and ROS-production machinery in

endosomes (Fig. 2, 3) are well performed and convincing.

The authors' further state that blocking of NOX2 activity suppresses TLR7 induced cytokine expression in vitro and in vivo. Additionally, inhibition of NOX2 enhances cytokine and antibody production during influenza infection (Fig. 4 and 5). These statements are well supported by the presented data.

Concerns

1) However, I feel that the explanation how ROS inhibits TLR7 falls a little short. Although the idea that Cys 98 functions as a redox sensor is very intriguing, the authors do not present data supporting this idea. The TLR7 mutant analysis (Fig. 4k) and the conservation of this residue in vertebrate TLR7 (supplement Fig.11) do not support its suggested role.

2) The idea of using NOX2 inhibition as a therapeutic strategy in influenza infection is quite new and exciting. However, I would suggest including a scrambled gp91phox sequence attached to the tat peptide as proper control for the in vivo experiments (e.g. cytokine production and viral burden (Fig6. 6i, m). Is it also possible to determine viral titers in the lung of infected mice (Fig. 6m)

Further points:

Figure 5 a,b: Standard deviation is missing

Figure 5b: y-axes values for IL-1 β and IL-6 production need a decimal place?

In summary, this manuscript is exciting, but the authors should address concern 1 and 2, before I would recommend publication of the manuscript.

Response to reviewer 1:

1. The legend to figure 1 is quite confusing, but as far as I understand, the authors show that bafilomycin inhibits the oxyburst signal and suggest that inhibition of endosomal acidification inhibits endosomal ROS generation by NOX2. Why would that be the case? How can the authors exclude that this is an artefact of the measurement technique.

The conclusion that reduced endosomal acidification inhibits ROS generation by NOX2 and the data in Figure 1 are strongly supported by significant publications demonstrating that reduced endosome acidification impairs the activation of TLR7 by viral RNA (Lund *et al.*, PNAS, 2004, 101 (15): 5598-5603; Diebold *et al.*, Science, 303(5663): 1529-1531).

This activation of endosome TLR7 by viral RNA then drives NOX2 activation and we have the following evidence to support this conclusion:

- a) We confirm that the inhibition of endosome acidification with Bafilomycin A (Fig 1G) abolishes the fluorescent signal for LysoTracker.
- b) Bafilomycin A suppressed the activation of TLR7 by the TLR7 agonist, imiquimod (Fig 4B). This shows that the activation pathway for TLR7 requires the low pH of the endosome.
- c) Bafilomycin A suppressed PKC activation due to influenza virus and imiquimod (Fig 2H). PKC is known to be upstream of acute NOX2 activation (for review of these pathways of NOX2 activation we refer to Bedard and Krause, *Physiol Rev.* 2007 Jan;87(1):245-313 and Drummond, *et al.*, 2011; *Nature Reviews Drug Discovery*).
- d) Bafilomycin A suppressed the association of p47phox-NOX2, which is a critical step for NOX2 assembly and activation (Fig 2E).

Therefore to fully address the reviewers comment we have modified the text to improve the discussion on the pH dependence of endosomal NOX2 ROS generation by the inclusion of the following sections in the manuscript:

Page 5 line 15-23: "We conclude that virus infection triggers a NOX2 oxidase-dependent production of ROS in endosomes using a process that is dependent on low pH. Indeed this conclusion is supported by the following experimental evidence. First it is known that reduced endosome acidification impairs the activation of TLR7 by viral RNA ^{18, 19}. Our study is in agreement with this finding, showing that NOX2 dependent ROS production to virus infection and to the TLR7 agonist imiquimod was abolished in TLR7^{-/-} cells and also by pretreatment with bafilomycin A. Second, bafilomycin A suppressed PKC activation due to influenza virus and imiquimod treatment, and PKC is upstream of acute NOX2 activation ^{10, 11}. Third, bafilomycin A suppressed the association of p47phox-NOX2, which is a critical step for NOX2 assembly and activation".

2. There are reasonable doubts about the quality and the specificity of the mouse NOX2 antibody used by the authors. Thus, the authors should show a very careful documentation of the quality and specificity of the antibody.

We have carefully chosen our NOX2 antibody, as we are aware of the potential issues with the quality of such products. However, please note that the antibody we used is the purified mouse anti-gp91phox (BD Transduction Laboratories, Purified Mouse Anti gp91[phox] Clone 53/gp91[phox](RUO) and not the AbCAM antibody that we indicated in our manuscript. We apologise for this typographical error, and understand the reviewers concern.

We have published evidence for the specificity of this NOX2 antibody (Fig 4 of Judkins *et al.*, 2010. 298, 1, H24-H32). In addition, to comply with the reviewers concern, we have performed a new experiment using this correct NOX2 antibody in alveolar macrophages taken from NOX2^{-/-} mice. We observed no staining in macrophages from the NOX2^{-/-} mice.

We have addressed the reviewers comment and our error by making the following changes to the manuscript.

Page 18; line 37: “purified mouse anti-gp91phox (BD Transduction Laboratories, Purified Mouse Anti gp91[phox] Clone 53/gp91[phox](RUO)”.

Page 14 line 34: “We verified the specificity of both the TLR7 and NOX2 antibodies by examining the degree of staining in WT macrophages and in TLR7^{-/-} and NOX2^{-/-} macrophages respectively. There was no staining for TLR7 in the TLR7^{-/-} macrophages (Fig 2c). Similarly with the NOX2 antibody we observed no staining in alveolar macrophages of NOX2^{-/-} mice compared to the WT cells (Supplementary Fig 1D). Further evidence for the specificity of this NOX2 antibody can be found in Judkins et al., 2010 ³²”.

A new panel has been added in supplementary Figure 1 (Supplementary Figure 1D) to show the staining of the NOX2 antibody in WT and NOX2^{-/-} alveolar macrophages.

3. The authors show that NOX2 activity reduces TLR-7 dependent signaling and hence there is an increased inflammatory response in NOX2-deficient phagocytes. The authors attribute this phenomenon to oxidation of cysteine residues in TLR-7. However the authors fail to acknowledge that the phenomenon of hyperinflammation in the absence of NOX2 is not at all specific for TLR-7. For example, very similar phenomena have been described in detail for fungal products and the activation of dectin-1 (e.g. J Pathol. 2008 Mar;214(4):434-44). Thus, the phenomenon of hyperinflammation is not TLR-7 specific and the proposed mechanism appears too simplistic.

This is an important point raised by this assessor and warrants further discussion in the manuscript. However, our study investigates the regulation of inflammation by subcellular, *endosomal* NOX2-ROS, which is critical to modulate viral infection, and for which endosomal TLR7 plays a pivotal role. Global NOX2 deficient phagocytes are likely to display consequences beyond the NOX2 expressed in endosomes because NOX2 is not only expressed in these compartments. And please note, we were not trying to state that hyperinflammation per se in the absence of NOX2 is entirely driven by the regulation of TLR7. However, from our data it appears that endosome NOX2-ROS is mainly regulating TLR7 to influence critical antiviral immunity and therefore inflammation to *virus* infection.

The study in J Pathology is very interesting, but it addresses the specific aspects of inflammation triggered by fungal components and the regulation of this inflammation by NOX2 oxidase. There is no evidence within that manuscript of negative regulation of any aspects of fungal signalling including the proposed pathways (in Figure 8 of that paper) involving TLR2. We addressed the potential role of TLR2 in our manuscript (Supplementary Figure S8). We found that TLR2KO macrophages displayed similar cytokine expression following catalase treatment compared to WT cells. Thus, it appears that H₂O₂ does not regulate TLR2 in a similar fashion as TLR7.

Given the importance of this point raised by the assessor, we have now incorporated a section in the discussion on other sources of ROS production.

See page 10 line 19: “Previous work has demonstrated that a deficiency in the phagocyte NADPH oxidase can result in exacerbated inflammation and necrosis in response to fungal components including β -glucans that activate TLR2 and dectin-1 ³⁰. Our study is not a universal mechanism that defines how NOX2 oxidase might be regulating inflammation to fungi and other pathogens but certainly provides strong mechanistic insight into ROS dependent regulation of antiviral immunity”.

4. I am also not fully convinced about the endosomal localization of the NOX2-dependent ROS generation. Indeed, the fact that the oxyburst measurement detects a signal does not prove that the ROS are generated in the endosomes. Indeed, H₂O₂ diffuses well and permeates membranes; therefore its site of generation and site of detection might be distinct.

This is acknowledged as an important point and we have therefore incorporated a new paragraph in the discussion to address this issue. We believe that the endosome will be a major source of ROS production in viral infection. Superoxide is the primary product of NOX2 and it will only be generated within the endosome compartment owing to the topology of the NOX2 and the unidirectional transfer of electrons through this catalytic subunit. In keeping with this, it is well regarded that superoxide

does not travel far from its site of generation due to its negative charge. By contrast to superoxide, hydrogen peroxide has some capacity to permeate membranes and diffuse and as such, it can be envisaged that some endosome H_2O_2 might have been generated elsewhere by NOX2 expressed in other sites of the cell such as by plasma membrane. However several lines of evidence suggest that very little remotely generated H_2O_2 is finding its way into the endosome compartment. First we demonstrate that PKC activation following virus infection, which is critical for NOX2 activation, is significantly impaired if: 1) the virus is prevented from entering cells, (Fig 2H and 2I); 2) endosome acidification is blocked by Bafilomycin A (Fig 2H and 2I) or 3) if TLR7 is absent (i.e. TLR7^{-/-} macrophages are used). Therefore, endosomal NOX2 derived ROS generation occurs *only* after virus has entered endosomes and activates endosome-specific pathways, lending further credence to endosome NOX2 as the predominant site of H_2O_2 generation. Second, elegant studies addressing spatial-temporal aspects of H_2O_2 diffusion very clearly demonstrate that H_2O_2 diffusion in the cytoplasm is strongly limited, but is instead localized to near the sites of its generation {Mishina, 2011 #40}, providing evidence that *endosome* H_2O_2 is unlikely to have been generated in remote sites.

We have made the following modifications to the manuscript to fully address the reviewers comment:

Page 9 line 31 to page 10 line 13: "We have accumulated evidence that virus entry into endosomal compartments triggers a NOX2 oxidase- dependent production of ROS in endosomes. However, some ROS may have possibly been generated in sites outside of the endosome, which may have then diffused into the endosomes. As such the site of ROS generation and site of detection in some instances might be distinct. We suggest that the major contributor to endosomal concentrations of superoxide will be superoxide generated directly in this compartment. Superoxide is the primary product of NOX2 and it will only be generated within the endosome compartment owing to the topology of the NOX2 and the unidirectional transfer of electrons through this catalytic subunit. In keeping with this, it is well regarded that superoxide does not travel far from its site of generation due to its negative charge. By contrast to superoxide, hydrogen peroxide has some capacity to permeate membranes and diffuse, and as such, it can be envisaged that some endosome H_2O_2 might have been generated elsewhere by NOX2 expressed in other sites of the cell such as the plasma membrane. However for several lines of evidence we suggest that it is very likely that little remotely generated H_2O_2 is finding its way into the endosome compartment. First we demonstrate that PKC activation following virus infection, which is critical for NOX2 activation, is significantly impaired if: 1) the virus is prevented from entering cells (Fig 2H and 2I); 2) endosome acidification is blocked by Bafilomycin A (Fig 2H and 2I) or 3) if TLR7 is absent (i.e. TLR7^{-/-} macrophages are used). Therefore, endosomal NOX2 derived ROS generation occurs only after virus has entered endosomes and activates endosome-specific pathways, lending further credence to endosome NOX2 as the predominant site of H_2O_2 generation. Second, elegant studies addressing spatial-temporal aspects of H_2O_2 diffusion clearly demonstrate that H_2O_2 diffusion in the cytoplasm is strongly limited, but is instead localized to near the sites of its generation ²⁹, providing evidence that endosome H_2O_2 is unlikely to have been generated at a remote site."

5. I see a similar problem with the endosomal NOX inhibitor Egp91 ds-tat. Indeed, the suggested site of action of the peptide inhibitor is on the cytosolic part of the NOX2 complex. Thus, the inhibitor needs to cross the endosomal (or plasma) membrane to efficiently inhibit. However once the compound has crossed the membrane, how can the authors be sure that it still acts locally around the endosome and does not diffuse towards other sites of NOX2 localization?

We would like to strongly emphasize that our novel NOX2 inhibitor is unlikely to solely suppress endosome NOX2. However, our inhibitor is specifically and preferentially delivered via the endocytic compartment owing to the cholesterol conjugation. In support of this, our findings of Figs 6A and B show that cholesterol conjugation results in a drug delivery system that promotes endosome delivery i.e our drug displayed a strong degree of co-location with EEA1+ endosomes that was blocked by dynasore pretreatment. This property is a novel delivery system that brings a NOX2 inhibitor to the predominant site of action that results from a virus infection (see Fig 6D showing strong co-location of viral nucleoprotein and our NOX2 inhibitor). Following internalization into the endosome the drug is likely to cross the endosomal membrane due to the TAT portion and suppress endosome NOX2 activity. The drug is likely to diffuse towards other sites of NOX2 localization, however, we believe the primary site will be endosomal NOX2 given that the drug is initially delivered via the endocytic pathway.

To clarify this issue made by the assessor we have made the following revision to the manuscript.

Page 9 line 16-28: "This is an innovative approach for suppressing NOX2 oxidase activity that occurs within the endosome compartment. We would like to emphasize that our novel NOX2 inhibitor is unlikely to solely suppress endosome NOX2. However, our inhibitor is specifically and preferentially delivered via the endocytic compartment owing to the cholestanol conjugation. In support of this, our findings of Figs 6A and B show that cholestanol conjugation results in a drug delivery system that promotes endosome delivery i.e our drug displayed a strong degree of co-location with EEA1+ endosomes that was abolished by dynasore pretreatment. This property is a novel delivery system that brings a NOX2 inhibitor to the predominant site of action that relates to virus infection (see Fig 6D showing strong co-location of viral nucleoprotein and our NOX2 inhibitor). We speculate that following internalization into the endosome the drug is most likely on the luminal face of the endosome membrane and due to the TAT portion can penetrate the membrane and suppress NOX2 activity. The drug might still be able to diffuse towards other sites or locations of NOX2, however, we believe the immediate and primary site of action will be NOX2 activity at the endosome, given that the drug appears to be selectively delivered via the endocytic pathway".

Response to reviewer 2:

1) I feel that the explanation how ROS inhibits TLR7 falls a little short. Although the idea that Cys 98 functions as a redox sensor is very intriguing, the authors do not present data supporting this idea. The TLR7 mutant analysis (Fig. 4k) and the conservation of this residue in vertebrate TLR7 (supplement Fig.11) do not support its suggested role.

This is an excellent point that has been raised by the assessor and while we anticipate that TLR7-Cys98 is redox regulated given that H₂O₂ is implicated, we feel that to prove this mechanism would require a dedicated piece of work, which is beyond the scope of this manuscript. Consequently, we have modified our conclusions on this point and added extra discussion to address the reviewers concern.

Page 7 line 39: "We suggest that H₂O₂ produced by endosomal NOX2 oxidase is likely to modify a single and evolutionary conserved unique cysteine residue i.e. Cys98 located on the endosomal face of TLR7, resulting in a dampened antiviral cytokine response. Potentially this signifies Cys98 of TLR7 as a novel redox sensor that controls immune function during viral infections. The precise type of molecular modification of this cysteine by H₂O₂, however, is currently unknown and certainly warrants further investigation."

2) The idea of using NOX2 inhibition as a therapeutic strategy in influenza infection is quite new and exciting. However, I would suggest including a scrambled gp91phox sequence attached to the tat peptide as proper control for the in vivo experiments (e.g. cytokine production and viral burden (Fig6. 6i, m). Is it also possible to determine viral titers in the lung of infected mice (Fig. 6m).

This is again an important point raised by the assessor. We were initially reluctant to perform such experiments, given the many publications demonstrating that the scrambled gp91phox sequence attached to the tat peptide fails to suppress ROS production.

It is clear from Figure 6H and 6I that the cholestanol conjugated gp91ds-TAT significantly reduced airways inflammation and increased IFN- β whereas the unconjugated gp91ds-TAT failed to do so. Thus one possible explanation for this improvement in potency might be related to conjugation per se, rather than the endosome targeting of NOX2 oxidase. For this reason, the scrambled control becomes critical as highlighted by the reviewer.

We have now generated a scrambled version of gp91ds-TAT and conjugated it to cholestanol via the PEG linker in the same manner as we did with the unscrambled drug. This scrambled version has been tested in a new series of *in vivo* experiments, as suggested by the reviewer. In the same protocol as Figure 6H, we treated mice with Sgp91ds-TAT (the term used for the scrambled version) at 0.02 mg/kg/day and measured airway inflammation and cytokine expression following influenza infection, as requested. Unlike Cgp91ds-TAT, the scrambled control had no effect on the airways inflammation and levels of Type I IFN following influenza virus infection. These data provide evidence

that the improvement in the effects of cholestanol conjugation of gp91dsTAT is not due to non-specific effects of cholestanol-PEG linker per se. We have created a new Supplementary Figure (i.e. Figure 13) to show this new data.

We have modified the manuscript in the follow manner:

Results section: Page 9 line 5: "To eliminate the possibility that this improvement in NOX2 inhibition by cholestanol conjugation of gp91ds-TAT was attributed to cholestanol-PEG linker per se, we conjugated the cholestanol PEG-linker to a scrambled gp91ds-TAT (Sgp91ds-TAT) and examined its effect against influenza infection in vivo. Sgp91ds-TAT had no effect on airway inflammation, lung IFN- β mRNA levels and superoxide production (Supplementary Fig 13)."

Methods section Page 12 line 30:

Sgp91 ds-tat: Ac-Asp(OChol)-PEG4-PEG3-PEG4-RKK-RRQRR-RCLRI-TRQSR-NH₂

Preparation of the 18 amino acid scrambled gp91 ds-tat (Sgp91 ds-tat) peptide was carried out by manual SPPS as described above for unscrambled gp91 ds-tat. The resin-bound sgp91 ds-tat was then conjugated to cholestanol via a PEG linker using the same method described above for unscrambled cgp91 ds-tat. The crude peptide was purified in the same way to give cgp91 ds-tat: calcd. for C₁₆₂H₃₀₇N₅₄O₄₀S [M + 5H⁺] m/z 736.3, obs. m/z 736.5; calcd. for C₁₆₂H₃₀₈N₅₄O₄₀S [M + 6H⁺] m/z 613.7, obs. m/z 614.0; calcd. for C₁₆₂H₃₀₉N₅₄O₄₀S [M + 7H⁺] m/z 526.2, obs. m/z 526.3.

Supplementary Section:

New figure 13.

3). Figure 5 a,b: Standard deviation is missing. This has now been corrected. Note, that all treatment groups are compared to the WT control group and expressed as fold-change, hence there is no error bar on the WT control group data.

4). Figure 5b: y-axes values for IL-1 β and IL-6 production need a decimal place? This has now been corrected.

REVIEWERS' COMMENTS:

Reviewer #2 (Remarks to the Author):

The manuscript is still very exciting and the additional experiments added further value to it. All my concerns were addressed adequately.

I would therefore suggest publishing this manuscript.